



# Late Cretaceous to Paleogene exhumation in Central Europe – localized inversion vs. large-scale domal uplift

Hilmar von Eynatten[1], Jonas Kley[2], István Dunkl[1], Veit-Enno Hoffmann[1], Annemarie Simon[1]

[1]University of Göttingen, Geoscience Center, Department of Sedimentology and Environmental Geology,
Goldschmidtstrasse 3, 37077 Göttingen, Germany
[2]University of Göttingen, Geoscience Center, Department of Structural Geology and Geodynamics,
Goldschmidtstrasse 3, 37077 Göttingen, Germany

*Correspondence to*: Hilmar von Eynatten (heynatt@gwdg.de)

**Abstract.** Large parts of Central Europe have experienced exhumation in Late Cretaceous to Paleogene time. Previous studies mainly focused on thrusted basement uplifts to unravel magnitude, processes and timing of exhumation. This study provides, for the first time, a comprehensive thermochronological dataset from mostly Permo-Triassic strata exposed adjacent to and between the basement uplifts in central Germany, comprising an area of at least some 250-300 km across. Results of apatite fission track and (U-Th)/He analyses on >100 new samples reveal that (i) km-scale exhumation affected the entire region, (ii) thrusting of basement blocks like the Harz Mountains and the Thuringian Forest focused in the Late Cretaceous (about 90-70 Ma) while superimposed domal uplift of central Germany is slightly younger (about 75-55 Ma), and (iii) large parts of the domal uplift experienced removal of 3 to 4 km of Mesozoic strata. Using spatial extent, magnitude and timing as constraints suggests that thrusting and crustal thickening alone can account for no more than half of the domal uplift. Most likely, dynamic topography caused by upwelling asthenosphere has contributed significantly to the observed pattern of exhumation in central Germany.

## 1 Introduction

Widespread intraplate compressional stresses affected Central Europe in Cretaceous to Paleogene time and generated numerous basement uplifts and inverted sedimentary basins (e.g. Ziegler et al., 1995; Kley and Voigt, 2008). The basement uplifts cover a large area of at least 1300 km west to east and 600 km north to south extension. It stretches from the Ardennes in Belgium (western Rhenish Massif) to south-eastern Poland (Holy Cross Mountains) and includes prominent fault-bounded blocks composed of crystalline basement rocks and pre-Permian metasedimentary rocks such as the Bohemian Massif, the Vosges and Black Forest, and the Harz Mountains (Fig. 1A). The major phase of exhumation and uplift is mostly assigned to the Late Cretaceous. However, earlier onset of exhumation and uplift and/or its continuation into the Paleogene are proposed for certain areas and structures (e.g. Barbarand et al., 2018; Sobczyk et al., 2020).







**Figure 1: (A) Pre-Tertiary geological sketch map of Central Europe (modified after Ziegler 1990). Black rectangle indicates position of the detailed geological map in Figure 3, straight line indicates the trace of the section shown in Figures 1B and 4. AR – Ardennes, FH – Flechtingen High, H – Harz Mountains, K – Karkonosze, NEGB – North-East German Basin, LSB – Lower Saxony Basin, MB – Münsterland Basin, OW – Odenwald, S – Sudetes, TB – Thuringian Basin, TF – Thuringian Forest, URG – Upper Rhine Graben. (B) Schematic geological section across the central part of the study area, highlighting the major, fault-bordered basement highs. (C) Compilation of apatite fission track ages obtained on structural highs exposing Paleozoic rocks in Central Europe: (a) Ibbenbüren High, Senglaub et al. (2005); (b) Flechtingen High, Fischer et al. (2012); (c) Harz Mountains, von Eynatten et al. (2019); (d) Halle volcanic complex, Jacobs and Breitkreuz (2003); (e) Northern Rhenish Massif, Karg et al. (2005); (f) Ardennes/Venn, Glasmacher et al. (1998), Xu et al. (2009) and references therein, Barbarand et al. (2018); (g) Thuringian Forest, Thomson and Zeh (2000); (h) Erzgebirge, Ventura and Lisker (2003), Lange et al. (2008), Wolff et al. (2015); (i) Lusatian Block, Lange et al. (2008), Ventura et al. (2009); (j) NE Bohemian Massif, Danisík et al. (2010), (2012), Migoń and Danišík (2012), Sobczyk et al. (2015), (2020); (k) Holy Cross Mountains, Botor et al. (2018); (l) E Bohemian Massif, Botor et al. (2017); (m) S Bohemian Massif, Hejl et al. (2003); (n) Barrandian, central Bohemian Massif, Glasmacher et al. (2002); (o) Bavarian Forest, Vamvaka et al. (2014); (p) W Bohemian Massif, Hejl et al. (1997); (q) Odenwald, Wagner (1968); (r) Black Forest/Vosges, Timar-Geng et al. (2006), Link (2009), Dresmann et al. (2010), Meyer et al. (2010).**

Inverted sedimentary basins (i.e., basins that have been exhumed along former extensional faults, cf. Cooper et al., 1989) occur partly between and within these blocks, and are common further north in the Central European Basin system (cf. Littke et al., 2008), which includes large parts of Poland, Northern Germany, The Netherlands, Denmark and the North Sea. The timing of basin inversion has been mostly assigned to Late Cretaceous to Paleogene time (e.g. Kockel, 2003; Krzywiec,

2006). Some authors have attributed all documented Mesozoic and Cenozoic uplift events in central Europe to increased tangential stress and inversion, regardless of their magnitude and extent (e.g. Ziegler et al., 1995; Sissingh, 2006). Others pointed out marked differences in the expression of these events and suggested that alternative mechanisms may be involved (Nielsen et al., 2005; Deckers and van der Voet, 2018; Kley, 2018).

This paper aims at a comprehensive understanding of the Late Mesozoic to Early Cenozoic exhumation in Central Europe

from a thermochronological point of view. We (i) review the existing thermochronological data on cooling and exhumation in Central Europe, (ii) present new thermochronological data from the main study area in the central part of Central Europe, (iii) integrate apatite fission track (AFT) and (U-Th)/He (AHe) data through thermal modelling that allows for estimating the thickness of eroded sequences, and (iv) discuss various models to explain the temporal and spatial pattern of exhumation and uplift in Central Europe.

**2 Geological Setting**

Central Europe has been an intraplate region since the Variscan orogeny that terminated about 300 Ma ago (e.g. Oncken, 1997). Its post-orogenic history began with the evolution and demise of the "Rotliegend" wide rift in Permian time (Lorenz and Nicholls, 1976). From the latest Permian through the Mesozoic a continuous cover of sediments was deposited over large parts of Central Europe. These sediments belong to the intracontinental Southern Permian Basin in the north (Littke et

al., 2008; Doornenbal and Stevenson, 2010) and to the proximal Tethys shelf in the south. Both basins were connected via an intervening platform and at times formed a contiguous region of marine deposition, e.g. in Middle Triassic (Muschelkalk)





and Early Jurassic time (Ziegler, 1990). The long-lasting slow subsidence in this system of basins was mostly of thermal origin (Cacace and Scheck-Wenderoth, 2016). Nevertheless, distributed and intermittent extension of generally low magnitude affected varying areas from the latest Permian to the Early Cretaceous (Geluk, 1999; Mohr et al., 2005; Warsitzka

et al., 2019). Despite the evidence for some large extensional faults (Best, 1996; Baldschuhn and Kockel, 1999; de Jager, 2003), normal fault offsets typically do not exceed a few hundred meters and there is no associated volcanism. Periods of intensified extension are recognized in Late Triassic (Keuper) time and, for our area, particularly in the Late Jurassic to Early Cretaceous when the Lower Saxony Basin formed (Ziegler, 1990; Stollhofen et al., 2008). Extension was centered on a belt stretching from the southern North Sea over the Netherlands and northern Germany to Poland. The southern border of this

belt is sometimes abrupt and sometimes gradual, involving normal faults at considerable distance from the main preserved depocenters (Kley et al., 2008; Danišík et al., 2012). Very subtle Mesozoic extension structures occur on the Helvetic shelf in the southwest (Wetzel et al., 2003; Malz et al., 2015).

The tectonic regime changed fundamentally from extension to contraction in Late Cretaceous time (Ziegler, 1987). Syntectonic basins formed along the margins of inverting sub-basins and uplifting basement blocks (Voigt, 1963; Krzywiec,

2002; Voigt et al., 2004; von Eynatten et al., 2008). The area of contraction closely coincides with the previous extension tectonics, even though not all major thrust faults are reactivated normal faults (Voigt et al., 2009). South of the main inversion axis extending from the North Sea Basins to the Polish Trough, contraction attenuates abruptly or gradually. For instance, shortening structures of small magnitude are widespread in the German uplands. Mesozoic structures in the Southern Permian Basin and the northern Alpine Molasse basin are sealed by an extensive cover of locally latest Cretaceous

but mostly Cenozoic sediments (Bachmann et al., 1987; Baldschuhn et al., 2001; Krzywiec and Stachowska, 2016; Voigt et al., this issue).

In Germany, uplift and erosion in Mesozoic and/or Cenozoic time are evidenced by large areas where the Variscan basement and Permian to Triassic strata are exposed today. The Rhenish and Bohemian massifs are commonly interpreted as long-lived highs that never had a substantial cover of Permo-Mesozoic sediments (Ziegler, 1990), although this model has been

recently challenged for parts of the Rhenish Massif (Augustsson et al., 2018). Between these massifs, Triassic strata were continuous from northern to southern Germany (Fig. 1A). For Jurassic and Cretaceous time much of the sedimentary record, if any, has been lost due to erosion in the central part of Germany (Fig. 1B). Remnants of Cenozoic strata show that denudation to the level of Triassic strata was completed by Paleogene or Neogene time, varying by region (Bundesanstalt für Geowissenschaften und Rohstoffe 1993).

Several processes have been proposed to have driven Late Cretaceous to Paleogene exhumation in Central and Western Europe. There is a consensus that Late Cretaceous inversion (often termed the "Subhercynian" event) was caused by far-field tectonic stresses related by different authors either to Alpine collision (e.g. Ziegler, 1987; Stackebrandt and Franzke, 1989; Ziegler et al., 1995; Krzywiec, 2006) or the onset of Africa-Iberia-Europe convergence (Kley and Voigt, 2008). In contrast, the Paleogene ('Laramide' and 'Pyrenean') uplift events have received very different interpretations. Many authors attributed

them to continued, if weaker, shortening and inversion (Ziegler, 1990; De Jager, 2003; Sissingh, 2006; Holford et al., 2009b)





or long-wavelength folding (Deckers and van der Voet, 2018). Nielsen et al. (2005) argued instead that the Laramide event reflects stress relaxation. This concept is consistent with numerical modelling suggesting that plate coupling across the Iberia-Europe boundary rapidly decreased with increasing incorporation of continental crust (Dielforder et al., 2019), but does not correctly predict the pattern of Laramide uplift and subsidence in some basins (Deckers and van der Voet, 2018).

Paleogene regional exhumation of the Irish Sea has been explained as isostatic response to magmatic underplating (Brodie and White, 1994; Ware and Turner, 2002). Kley (2018), based on its very large areal extent, advocated dynamic topography and lithospheric thinning (see also Meier et al., 2016) as the causes of Laramide uplift. In this paper we focus on the timing, magnitude and possible mechanisms of Late Cretaceous and Paleogene exhumation in Germany. Because of the coincidences in time with widespread exhumation events in Central Europe, these will be reviewed in the next section.

## 95 3 Review of thermochronological data

The available thermochronological data on Mesozoic to Tertiary exhumation in Central Europe are widespread and focus on individual regions exposing Paleozoic basement rocks. Only near-surface samples are considered for this compilation to ensure comparability. The most comprehensive data set is available for apatite fission track data, which are summarized in Figure 1C with respect to individual regions and the respective range of AFT ages. Many regions show predominance of

Cretaceous AFT ages, however, others show a much broader range including Jurassic and Permo-Triassic ages and/or significant contribution of Tertiary ages.

The Ardennes, forming large part of the western Rhenish Massif in the Rhenohercynian zone of the Central European Variscan orogeny (location *f* in Fig. 1C), are predominantly composed of very low-grade Devonian to Carboniferous metasedimentary rocks, with some Early Paleozoic low to medium grade inliers. Low-T thermochronology comprise mostly

AFT data along with some zircon fission track (ZFT) data. The latter indicate ages between 422 and 218 Ma, while AFT ages range from Permian to Early Cretaceous (290 to 130 Ma; Glasmacher et al., 1998; Xu et al., 2009 and references therein; Barbarand et al., 2018). Although the age range is rather consistent across the various studies, their interpretations based on thermal modelling and various geological evidence are quite different. Xu et al. (2009) propose slow exhumation and cooling for most of the Mesozoic followed by accelerated cooling since Mid-Eocene time. Glasmacher et al. (1998)

suggest a major phase of cooling in the Mid-Cretaceous (120 to 80 Ma) along with approx. 2000 m of exhumation. In contrast, Barbarand et al. (2018) use paleoweathering geochronology based on Mn-oxide phases to constrain the thermochronological modelling. They suggest Late Jurassic to Early Cretaceous uplift and erosion of the Ardennes Massif, which subsequently stays close to the surface until present. For the northeastern Rhenish Massif, on the left side of the River Rhine, Karg et al. (2005) reported AFT ages ranging from 291 to 136 Ma (location *e* in Fig. 1C). The data indicate cooling in

the late stages and/or after the Variscan orogeny. Triassic to Jurassic sedimentation is interpreted below 1000 m in thickness and the samples remained in the apatite PAZ during most of the Mesozoic. Final uplift and denudation did not start before Late Cretaceous, with slightly accelerated cooling in the Tertiary (Karg et al., 2005). Similar data have already been reported



by Büker (1996) for the same area (AFT 286–159 Ma). No thermochronological data have been published so far from the eastern margin of the Rhenish Massif, however, some new data will be presented in this study.

East to northeast of the Rhenish Massif, several thrusted basement blocks expose Paleozoic crystalline and/or metasedimentary rocks at the surface. These are the Ibbenbüren High, the Flechtingen High, the Harz Mountains and the Thuringian Forest (*a*, *b*, *c* and *g* in Fig. 1C, respectively). They are all characterized by long apatite fission tracks and narrow track length distributions, and despite the wide area and a pretty high number of samples (n = 41) the AFT age range is impressively tight, i.e. fully restricted to Campanian time (83-72 Ma; Thomson and Zeh, 2000; Senglaub et al., 2005; Fischer

et al., 2012; von Eynatten et al., 2019). For the Harz Mountains, the observation of rapid Late Cretaceous exhumation is further supported by zircon (U-Th)/He (ZHe) and AHe thermochronology (von Eynatten et al., 2019) as well as independent evidence from facies distribution and provenance information from well-dated syntectonic sediments in the foreland (Voigt et al., 2006; von Eynatten et al., 2008). ZFT data from the four basement blocks range from the latest Carboniferous (306 Ma) to latest Triassic (202 Ma). While the oldest ages may directly reflect cooling after the Variscan orogeny, the bunch of

Permo-Triassic ages are interpreted as mixed ages, reflecting partial reset due to widespread Permo-Mesozoic burial heating (Fischer et al., 2012; von Eynatten et al., 2019). Slightly east of the Harz Mountains, the Halle volcanic complex (location *d* in Fig. 1C) consists of Permo-Carboniferous felsic volcanic rocks yielding thermochronological data rather similar to the Harz Mountains, although AFT ages are slightly older (ZFT 231-194 Ma and AFT 108-74 Ma; Jacobs and Breitkreuz, 2003). The Bohemian Massif represents the largest inlier of basement rocks exposed in Central Europe (Linnemann et al., 2008). It

is mainly composed of Late Neoproterozoic to Early Paleozoic (Cadomian) and Late Paleozoic (Variscan) granitoids and a large variety of metamorphic rocks comprising Late Neoproterozoic to Carboniferous protoliths and metamorphic grades from very-low grade to high-grade including ultrahigh-pressure metamorphism (Kroner et al., 2008; Schönig et al., 2020). Morphologically, the Bohemian Massif comprises several mountain ranges. Their low temperature evolution has been investigated by numerous studies in order to unravel the post-Variscan exhumation and uplift history of the Bohemian

Massif. Results indicate a complex spatial pattern caused by various partly superimposed processes including Mesozoic burial, Late Cretaceous exhumation due to far field compression and/or reheating and exhumation related to the European Cenozoic Rift System and associated volcanism (e.g. Danišík et al., 2012).

The northern margin of the Bohemian Massif can be roughly separated, from west to east, into the Erzgebirge, the Lusatian Block and the Sudetes including the Karkonosze Mountains. The central Erzgebirge is characterized by medium to high

grade metamorphic rocks including relics of ultrahigh-pressure metamorphism, surrounded by lower-grade metasedimentary rocks and intruded by late Variscan granitoids. AFT data reveal a large age spread from 210 to 45 Ma with predominantly (19 out of 24) Late Jurassic and Cretaceous ages (location *h* in Fig. 1C and Fig. 2A; Lange et al., 2008). Similar ages with similar median value are reported from a borehole in the western Erzgebirge (Ventura and Lisker, 2003). The diverse spatial pattern of thermochronological data is supported by ZHe and AHe data and indicates dissection into individual structural

blocks, variable Mesozoic burial and hydrothermal activity, significant Late Cretaceous exhumation and only minor Cenozoic overprint (Wolff et al., 2015). The Lusatian block (location *i* in Fig. 1C), which is separated from the Erzgebirge





by the Elbe fault zone, is characterized by mostly Late Cretaceous AFT ages, as reported by two different studies. The first is based on 28 predominantly granodiorite samples covering the entire area, which yield an AFT age range of 102 to 50 Ma (Lange et al., 2008). The second is based on 10 samples from three boreholes (sample depth <500 m) from the northern and
southern boundaries of the block, which yield a tighter Late Cretaceous AFT age range from 94 to 72 Ma (Ventura et al., 2009). Roughly similar data are reported from various studies of the Sudetes, forming the structurally complex NE margin of the Bohemian Massif (location *j* in Fig. 1C). The Variscan granite intrusion of the Karkonosze Mountains experienced punctuated Late Cretaceous fast cooling and exhumation as evidenced by ZHe (98 Ma), AFT (90-82 Ma) and AHe (87-79 Ma) thermochronology (Danišík et al., 2010). Metamorphic rocks to the north of the intrusion show a somewhat broader
Cretaceous age range (121-63 Ma; Martínek et al., 2006; 2008, cit. in Migoń and Danišík, 2012). From the easternmost part of the intrusion and adjacent metamorphic rocks Early Cretaceous ZHe ages (131-100 Ma) and Cretaceous to Paleogene AFT ages (106-51 Ma) are reported (Sobczyk et al., 2015). In the SE Sudetes, crystalline rocks of the Rychlebské hory Mountain region reveal Late Cretaceous to Paleocene exhumation history evidenced by ZHe (79 and 89 Ma) and AHe (90-69 Ma) data along with slightly younger AFT ages (81-39 Ma), the youngest of them were most likely influenced by
Cenozoic volcanism (Danišík et al., 2012). However, nearby samples from small crystalline massifs reveal similar Late Cretaceous to Eocene AFT ages (84-45 Ma), which are interpreted to reflect Late Cretaceous onset of cooling with a climax in the Paleocene to Middle Eocene (Sobczyk et al., 2020). The cumulative distribution of all AFT ages from the NE margin of the Bohemian Massif shows significant age groups in the Late Cretaceous, around the K-T boundary and in the Paleocene to Eocene (Fig. 2B).
At the eastern margin of the Bohemian Massif, the Moravo-Silesian zone exposes Lower Carboniferous synorogenic clastic sedimentary rocks (Moravo-Silesian Culm Basin, location *l* in Fig. 1C), which experienced post-Variscan anchimetamorphic overprint. ZHe ages range from 303 to 163 Ma, while AFT ages range from Late Jurassic (152 Ma) to Eocene (44 Ma) with the majority of ages belonging to the Late Cretaceous (Fig. 2C; Botor et al., 2017). In the southernmost part of the Bohemian Massif (Waldviertel) AFT ages from granitoids and gneisses show a broad range from 233 to 92 Ma (location *m* in Fig. 1C;
Hejl et al., 2003). Despite some spatial variation and generally older ages towards the northeast, thermal modelling suggests general Jurassic to Early Cretaceous cooling, followed by some reburial in the Late Cretaceous. Cenozoic bulk denudation is estimated in the order of 1-3 km (Hejl et al., 2003). At the southwestern margin of the Bohemian Massif, the crystalline rocks of the Bavarian Forest (location *o* in Fig. 1C) yield AFT ages that are almost entirely Cretaceous (148-83 Ma) with most of them falling into the Late Cretaceous (median 93 Ma, Fig. 2D; Vamvaka et al., 2014). Thermal modelling of these
data suggests enhanced heat flow in the Middle Jurassic to Early Cretaceous, likely caused by lithospheric extension, followed by enhanced Late Cretaceous to Paleogene cooling. At the western margin of the Bohemian Massif (Oberpfalz; location *p* in Fig. 1C), Permo-Triassic ZFT ages (283-215 Ma) are thought to indicate post-Variscan unroofing and denudation (Hejl et al., 1997). This observation is complemented by a series of AFT data that reveal a relatively tight age cluster (110-54 Ma) implying accelerated denudation in Late Cretaceous to Paleogene time (Fig. 2D).






**Figure 2: Cumulative frequency distributions of apatite fission track data from various regions across Central Europe (references according to Figure 1). Characters in brackets refer to location in Figure 1. Bold numbers indicate the median ages of the distributions. Green area indicates the time interval of the Late Cretaceous.**





In contrast to the numerous studies from the margins of the Bohemian Massif reporting Cretaceous AFT median ages (Fig. 2A-D), data from the Tepla-Barrandian and adjacent areas in the center of the Bohemian Massif are relatively scarce and indicate significantly older AFT ages (324-161 Ma, location *n* in Fig. 1), which are interpreted to indicate late Variscan cooling, reburial during the post-Variscan Molasse stage and relatively slow Mesozoic cooling (Glasmacher et al., 2002; Suchý et al., 2019).

Northeast of the Bohemian Massif, the Holy Cross Mountains in SE Poland (location *k* in Fig. 1C) exposes Paleozoic sedimentary rocks that experienced deformation during the Variscan orogeny, followed by significant burial during prolonged Permo-Mesozoic subsidence within the Mid-Polish Trough of the Polish Basin and exhumation in the Late Cretaceous to Paleocene (e.g. Krzywiec et al., 2009). The available thermochronological data reveal mostly Late Paleozoic ZHe ages (417-283 Ma) that are generally younger than the sedimentation ages, along with few scattered Mesozoic AFT ages (202-88 Ma) and Late Cretaceous to Paleogene AHe ages (91-43 Ma). Thermal modelling indicates late to post-Variscan (Carboniferous to Permian) cooling from maximum deep diagenetic temperatures reached during the Variscan orogeny. Mesozoic burial and Late Cretaceous fast cooling and exhumation is most pronounced in the eastern part of the Holy Cross Mountains (Botor et al., 2018).

The Upper Rhine Graben (URG) forms part of the Cenozoic European Rift system and exposes crystalline basement domains at its exhumed flanks. The most prominent ones are located adjacent to the southern URG, i.e. the Vosges at the western and the Black Forest at the eastern flank (location *r* in Fig. 1C). Thermochronological data show broad age ranges, reflecting the complex superposition of variable Mesozoic hydrothermal activity (that reached the zircon partial annealing zone most likely in Jurassic time), Late Cretaceous cooling and exhumation, and Cenozoic re-heating related to rifting in the URG followed by final exhumation (Timar Geng et al., 2006; Link, 2009; Dresmann et al., 2010; Walter et al., 2018). ZFT ages range from Late Carboniferous to Early Cretaceous (312-109 Ma; the youngest ages are related to a fault zone at the southern margin of the Black Forest, Dresmann et al., 2010), while AFT ages range from Cretaceous to Miocene (103-15 Ma) with significant age components in the Late Cretaceous and Paleogene (Fig. 2E). AHe data yield only Cenozoic ages ranging from Paleocene to Early Miocene (61-20 Ma; Link, 2009). Adjacent to the northern URG, the Odenwald region (location *q* in Fig. 1C) exposes Variscan granitoids from the Mid-German Crystalline Rise, which yield a relatively tight Cretaceous AFT age range (105-70 Ma, Fig. 2F; Wagner, 1968). Late Cretaceous AFT ages (~80-70 Ma) of the Odenwald as well as the opposed western flank of the URG (Palatinate Forest) are confirmed by Link (2009). The almost entire restriction to Late Cretaceous AFT ages point to an essentially similar thermal evolution as described before for the Thuringian Forest and the Harz Mountains further northeast (Fig. 1).

In summary, thermochronological data from most of the basement uplifts in Central Europe provide evidence for a phase of Late Cretaceous cooling and exhumation. This is most obvious and accentuated for an approximately 300 x 250 km region in central Germany, encompassing the Harz Mountains and the Thuringian Forest, as well as the Flechtingen and Ibbenbüren Highs in the north and the Odenwald in the south (i.e. the main study area as outlined in section 4). Further south, at the





flanks of the URG, this event seems also important but is partly masked by Cenozoic rifting and magmatic-hydrothermal activity. Towards east, the margins of the Bohemian Massif provide multiple evidence for Late Cretaceous cooling and exhumation, with variable expansion into the Paleogene, especially at its western and eastern margins. Even for the most remote areas in the west (Ardennes) and east (Holy Cross Mts.), Late Cretaceous to Paleogene cooling is considered a

relevant part of the post-Variscan evolution.

## 4 Study area in central Germany and sample coverage

The main study area approximately coincides with the northern half of the German uplands region ('Mittelgebirge', Fig. 3). It is underlain by a mostly sedimentary substrate of variable lithology and structural complexity. Its northern part borders on the inverted Lower Saxony Basin (LSB) in the west and the Northeast German Basin (NEGB) in the east (Fig. 1). The

southern border of the LSB is thrust onto the Cretaceous Münsterland Basin along the Osning Fault, a regional reactivated normal fault. The Münsterland Basin overlies the margin of the Rhenish Massif. The NEGB is bounded to the south by the Gardelegen reverse fault which creates a major basement step (Fig. 4). Two additional prominent basement uplifts related to major Cretaceous thrust faults follow towards the south (Harz Mountains and Thuringian Forest), separated by wide synclinal areas dominated by Triassic strata. The Gardelegen Fault and northern Harz thrust fault are associated with syn-

inversion Cretaceous growth strata in the NEGB and the Subhercynian Basin, respectively. The Thuringian Forest grades into the Thuringian Schiefergebirge and the Bohemian Massif towards the southeast. All three are bordered by the Franconian Line, a regional southwest-directed reverse fault. The Osning Fault and Franconian Line approximately mark the southwestern limit of strong Mesozoic deformation. They are connected by an array of smaller fault zones (e.g. the Hessian grabens), most of which are extensional structures displaying signs of inversion. Similar fault zones south of the Franconian

Line are the southernmost expression of Late Cretaceous inversion on the transect of Fig. 4 (Kämmlein et al., 2020). The Odenwald and Spessart in the west are uplifts of the Variscan basement not associated with major Cretaceous thrust faults (Fig. 3). The gently warped Franconian platform underlies the cuesta landscape of southern Germany ('Süddeutsches Schichtstufenland') and dips under the Cenozoic of the northern Alpine Molasse foreland basin in the south.





**Figure 3: Geological map of the study area (simplified after Geowissenschaftliche Karte der Bundesrepublik Deutschland 1: 2 000 000, Bundesanstalt für Geowissenschaften und Rohstoffe, Hannover (2004); \* affected by variable degree of metamorphic overprint). Abbreviations are used for geological-structural units mentioned in the text and correspond to the stratigraphical columns in Fig. 5a: LSB – Lower Saxony Basin, HFB – Harz Foreland Basin, HM – Harz Mountains, LB – Leipzig Basin, TB – Thuringian Basin, TF – Thuringian Forest, WFW – Werra-Fulda-Weser region, E-RM – Eastern Rhenish Massif, FP – Franconian Platform, W-BM – Western Bohemian Massif. Locations of further geographical-geological units mentioned in the text are labelled with numbers: 1 – Thuringian Schiefergebirge, 2 – Odenwald, 3 – Spessart, 4 – Vogelsberg, 5 – Rhön, 6 – Kyffhäuser High, 7 – Münsterland Basin.**






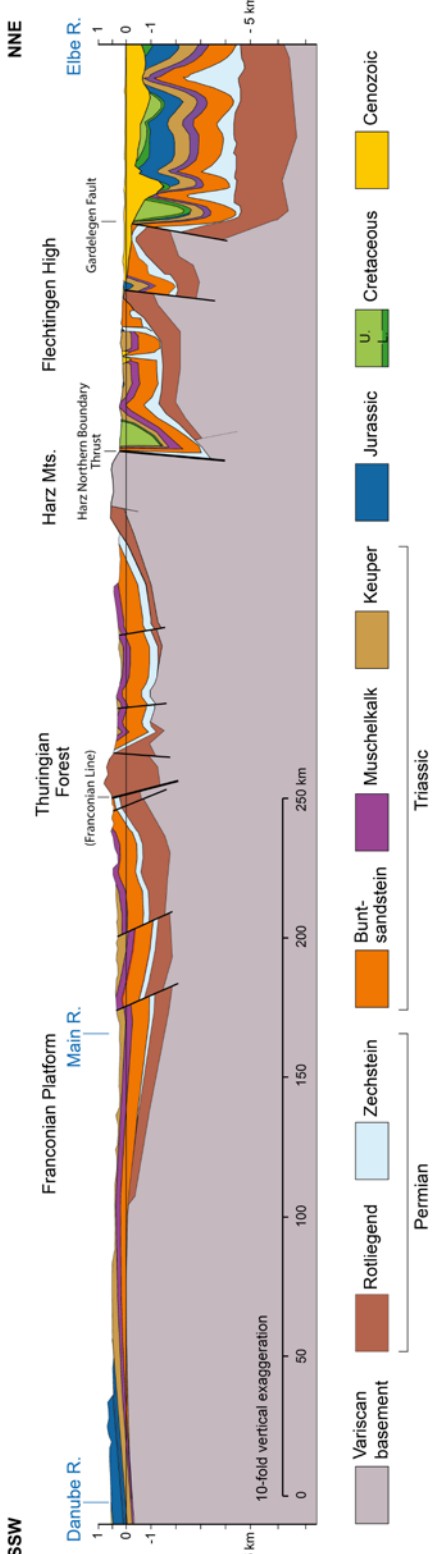

**Figure 4: Regional cross-section from the Danube to the Elbe (location in Figure 1). Strong inversion-related deformation from the Thuringian Forest towards the north contrasts with long-wavelength undulations of the area to the south. Faults are shown schematically but with correct general dip. Area north of the Harz Mountains is simplified from Jähne (in Kley et al., 2008, p. 105) and Malz et al. (2020).**



Ten major geological-structural units are defined in the study area considering the major fault patterns as well as contrasts in the thickness, facies, and preservation of the Mesozoic-Cenozoic sequences (Fig. 5). Only the Lower Saxony Basin and the

Harz Foreland Basin show a more-or-less complete record of Mesozoic sedimentation. In the central part of the study area, drained by the Werra, Fulda and Weser rivers (WFW), as well as the Thuringian Basin and the Franconian Platform, Triassic sedimentary rocks are exposed at the surface while Jurassic and Cretaceous strata are rarely preserved (Fig. 3). The latter applies also for the Leipzig Basin, but there a largely continuous Tertiary cover is preserved. The deepest denudation has obviously affected the crystalline basement highs, namely the Harz Mountains, Thuringian Forest, Rhenish Massif and

Bohemian Massif, where Variscan metamorphic and intrusive rocks and some Permian deposits are exposed at the surface and no stratigraphic evidences are available on the thickness of the former Mesozoic and Cenozoic cover.

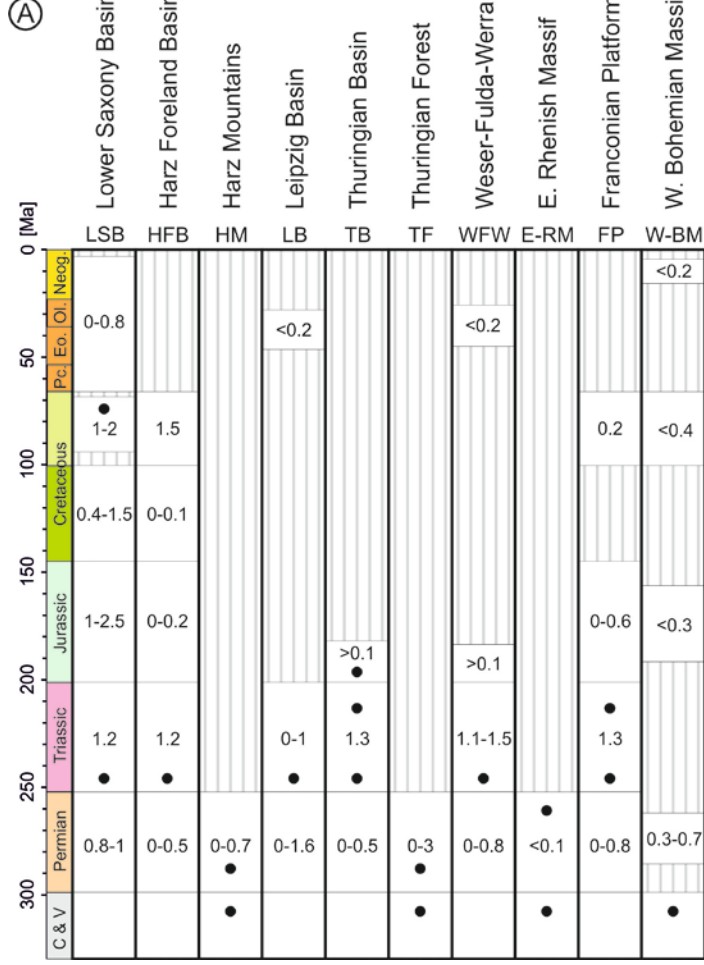



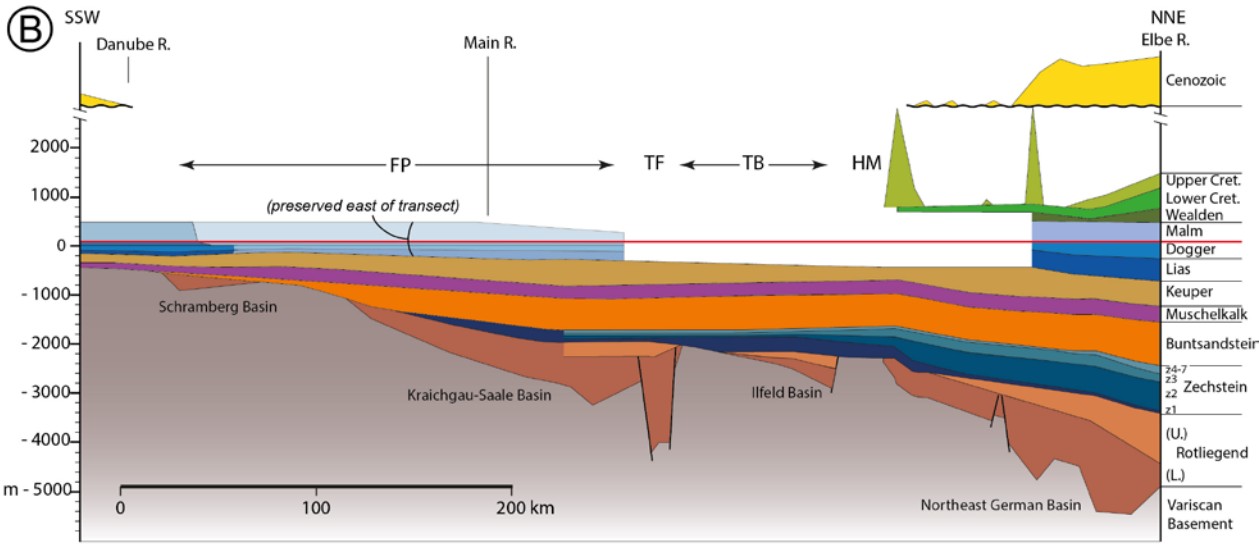

**Figure 5: (A) Compilation of stratigraphical data for various geological-structural units of the study area as mentioned in the text and indicated in Figure 3. The approximate thicknesses of the strata are given in km. C & V: Carboniferous (meta)sedimentary or Variscan metamorphic or igneous basement rocks. Black dots represent the stratigraphic levels of the samples used for thermochronology. (B) Transect along the cross section in Figure 4, showing primary variations in thickness and different preservation. The units shown in (A) are projected onto the transect if appropriate. Zechstein to Triassic strata exhibit a relatively monotonous southward thickness decrease and onlap onto Variscan basement of the southern basin margin. Eroded thicknesses of Jurassic and Cretaceous strata from the central segment are primarily constrained by thermochronological data. Thickness data are from Boigk and Schöneich (1974), Hoth et al. (1993), contour maps in Freudenberger and Schwerd (1996) and Franke (2020).**

## 4.1 Samples

More than 300 samples have been collected and the results are based on the analysis of more than hundred samples suitable

for thermochronology, covering an area of about 300 km in E-W extension and 220 km in N-S extension (see Supplement S1 and S2). Because the basement uplifts are largely well characterized in terms of thermochronology (section 3), the sampling strategy focuses on areas in between these uplifts such as the Thuringian Basin, the Weser hills (Weserbergland) and the Hessian Grabens as well as areas marginal to the Rhenish Massif and the western Bohemian Massif. Moreover, a couple of samples have been taken from the Thuringian Forest and the Franconian platform to trace the thermal history of the Triassic

strata towards south.

Samples comprise mostly siliciclastic rocks from the Early Triassic (Buntsandstein, n=63), Late Paleozoic (Devonian to Permian, n=33), Late Triassic (Keuper, n=8) and Early Jurassic (n=1). Further samples include Variscan granite (n=2 including one granite pebble from Permian clastics), Devonian diabase (n=2) and Permian rhyolite (n=1). The distribution of the samples with respect to stratigraphical levels of the different geological-structural units in the study area is outlined in

Figure 5A. Almost all samples are surface samples except for four drillcore samples, taken from depths shallower than 500





m. All of the 110 samples listed were used for apatite fission track analysis, while 37 of them were also suited for apatite (U-Th)/He thermochronology.

## 5 New thermochronological data

### 5.1 Methods

For *apatite fission track analysis* the external detector method was used (Gleadow, 1981). Highly enriched apatite concentrates were embedded in epoxy resin, diamond polished in five steps and etched by 5.5 N $HNO_3$ solution for 20 sec at 21°C (Donelick et al., 1999). The apatite grain mounts with the etched spontaneous tracks were covered with freshly cleaved muscovite sheets as external track detectors and irradiated with thermal neutrons in the research reactor of the TU Munich in Garching. Corning glass dosimeter (CN5) was used to monitor the neutron fluence. After irradiation the tracks in the external

detectors were revealed by etching in 40% HF for 40 min at 21°C. Spontaneous and induced fission tracks were counted under 1000x magnification using a Zeiss Axioskop microscope equipped with computer-controlled stage system (Dumitru, 1993). Only apatite crystals with well polished surface parallel to the crystallographic c axis were considered. In most cases 20 to 25 grains were measured per sample. Additionally the Dpar values were measured in each dated apatite crystal and the lengths of horizontal confined tracks were determined in most of the samples. AFT ages were calculated using the zeta age

calibration method (Hurford and Green, 1983) with the standards listed in Hurford (1998). Data processing and plotting were performed with the TRACKKEY software (Dunkl, 2002) while errors were calculated using the classical procedure described in Green (1981).

For *apatite (U-Th)/He analysis* the crystals were wrapped in platinum capsules and heated in the full-metal extraction line by an infra-red laser for 2 minutes in high vacuum. The extracted gas was purified using a SAES Ti-Zr getter kept at 450 °C.

The chemically inert noble gases and a minor amount of other rest gases were then expanded into a Hiden triple-filter quadrupol mass spectrometer equipped with a positive ion counting detector. Beyond the detection of helium the partial pressures of some rest gases were continuously monitored ($H_2$, $CH_4$, $H_2O$, $N_2$, Ar and $CO_2$). He blanks were estimated using the same procedure on empty Pt tubes and the crystals were checked for degassing of He by sequential reheating and He measurement. The amount of He extracted in the second runs are usually below 1%. Following degassing, samples were

retrieved from the gas extraction line, spiked with calibrated $^{230}$Th and $^{233}$U solutions. The apatite crystals were dissolved in a 4% $HNO_3$ + 0.05% HF acid mixture in Savillex teflon vials. Each sample batch was prepared with a series of procedural blanks and spiked normals to check the purity and calibration of the reagents and spikes. Spiked solutions were analyzed by a Thermo iCAP-Q ICP-MS. The ejection correction factors (Ft) were determined for the single crystals by a modified algorithm of Farley et al. (1996) using an in-house spreadsheet.

Single-crystal aliquots were dated, usually three aliquots per sample. The crystals were inspected for inclusions under 250x magnification and cross-polarized light. Inclusion- and fissure-free, intact, mostly euhedral apatite crystals were selected from igneous samples. The siliciclastic samples, however, contain mostly well-rounded apatite grains with dull or rugged





surfaces. The interiors of such grains were controlled in alcohol immersion, but due to the limited optical resolution they may contain tiny inclusions, that were overlooked and may have led to minor contributions of excess helium. The shape
parameters for the Ft-correction were determined and archived by multiple digital microphotographs. The morphology of the euhedral crystals was approximated by the combination of prismatic and pyramidal faces, while in case of rounded grains it was approximated by oblate and prolate ellipsoids.

## 5.2 Results of apatite fission track analysis

The AFT apparent ages range from 208 to 53 Ma with the majority of ages (~60%) falling into the Late Cretaceous (Fig. 6
and Supplement S3). Paleocene to earliest Eocene ages contribute with ~21% and Early Cretaceous ages with ~17% to the entire distribution while older ages are very rare. Track length measurements reveal mean values between 11.5 and 14.1 µm and standard deviations between 0.9 and 2.7 µm (Supplement S4). Relations between track length data and apparent ages display boomerang-like shapes (Green, 1986), i.e. the younger and the oldest ages tend to have slightly longer tracks and narrow track length distributions (Fig. 6). Dpar values range from 1.70 to 3.47 µm with a mean of 2.46 µm, pointing to
overall relatively high thermal retentivity of fission tracks in the lattice of the dated apatite crystals (Donelick et al., 2005).

For evaluation, the new apatite fission track data are assigned to five regions according to the spatial clustering of the samples, data coherence, and their belonging to the geological-structural units and stratigraphic levels as introduced in Figs. 3 and 4. These are (i) the core of the study area formed by the Buntsandstein uplands in Northern Hesse and southern Lower Saxony, drained by the Weser, Fulda and Werra rivers (WFW), (ii) Triassic of the center of the Thuringian Basin (TB), (iii)
the transition from the eastern Thuringian Basin into the Thuringian Schiefergebirge (i.e. western margin of the Bohemian Massif; W-BM), (iv) Late Paleozoic to Early Triassic of the eastern margin of the Rhenish Massif (E-RM) and (v) the Franconian Platform (FP) south of the Thuringian Forest (Fig. 7). The elevated basement highs of the Harz Mountains (HM) and the Thuringian Forest (TF) yield also very coherent age data. For the evaluation of these two units a few new data are combined with published data by von Eynatten et al. (2019) and Thomson and Zeh (2000), respectively. Note that not all
samples are included in these seven regions.

The core region of the study area, WFW, exhibits Late Cretaceous to earliest Eocene ages (90–53 Ma). It shows a relatively large contribution of early Tertiary ages (17 out of 26) compared to the overall age distribution and includes the youngest ages of the entire study area (Fig. 7A). South of the Thuringian Forest and on the Franconian platform (FP) AFT data reveal a narrow Late Cretaceous to Paleocene AFT age range (74–57 Ma), rather similar to WFW. In the Thuringian Basin AFT
apparent ages range from 123 to 57 Ma. Late Cretaceous ages are predominant in the central part of the basin (TB, except for one Paleocene age); however, towards its eastern margin several Early Cretaceous ages are observed (123–104 Ma). This trend towards older ages extends into the Thuringian Schiefergebirge with Early Cretaceous to Late Jurassic ages (151–131 Ma). Because this trend does not allow clear separation the eastern Thuringian Basin and the Schiefergebirge have been grouped together (W-BM; Fig. 7A). A similar situation, although flipped in the orientation of the age trend, is observed at
the eastern margin of the Rhenish Massif (E-RM). Late Paleozoic rocks within the massif reveal, besides Late Cretaceous





ages, several Late Jurassic to Early Cretaceous ages (101–151 Ma) and a single Late Triassic age at the westernmost location. East of the Rhenish Massif, Early Triassic sandstones reveal exclusively Late Cretaceous ages (90–67 Ma; Fig. 7A). The Permo-Triassic samples clustering to the southeast of the Harz Mountains and around the fault-bounded Kyffhäuser High range between 106 and 61 Ma. Due to structural complexity they were not treated as a coherent group here.

The oldest age (106) Ma occur at the western edge of the Kyffhäuser High and suggests transition to the Early Cretaceous ages observed along the western and southern rim of the Harz (von Eynatten et al., 2019).

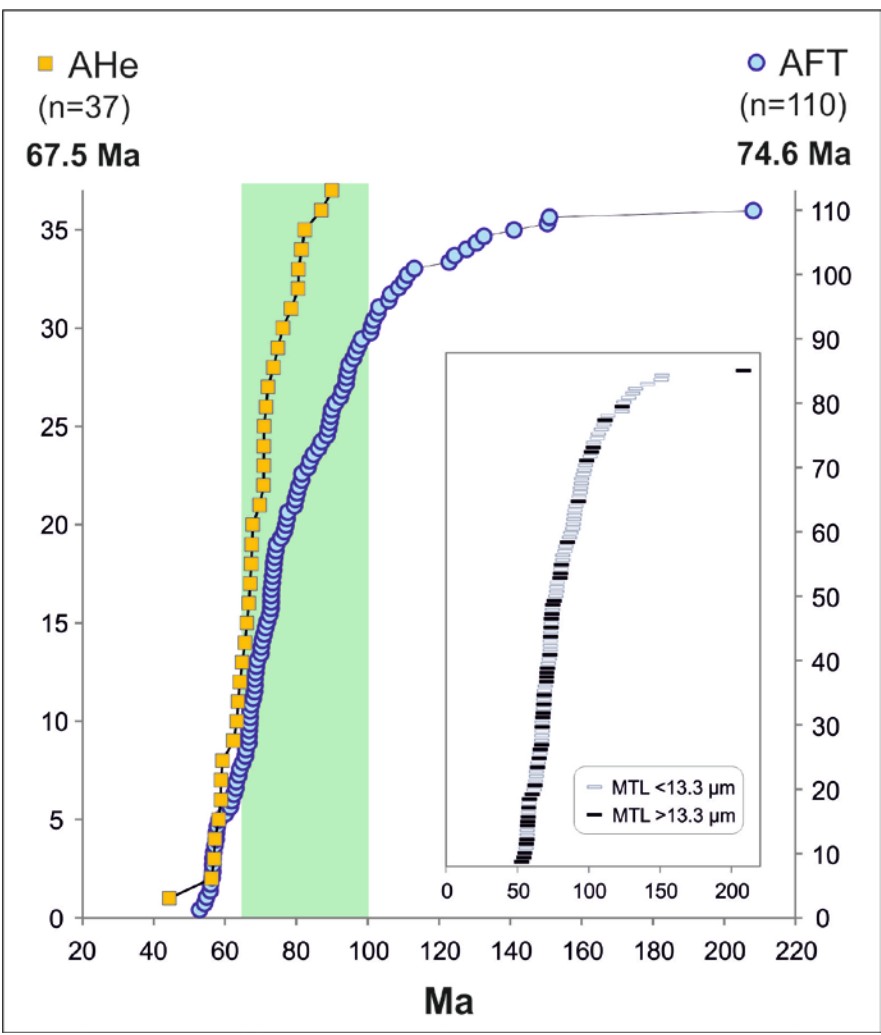

**Figure 6: Cumulative frequency distributions of apatite fission track (AFT) and apatite (U-Th)/He (AHe) ages from the study area. Bold numbers indicate the medians of the two data sets. Green area indicates the time interval of Late Cretaceous. The inset highlights that long mean track lengths (MTL) are mostly associated with AFT ages younger than ~80 Ma.**





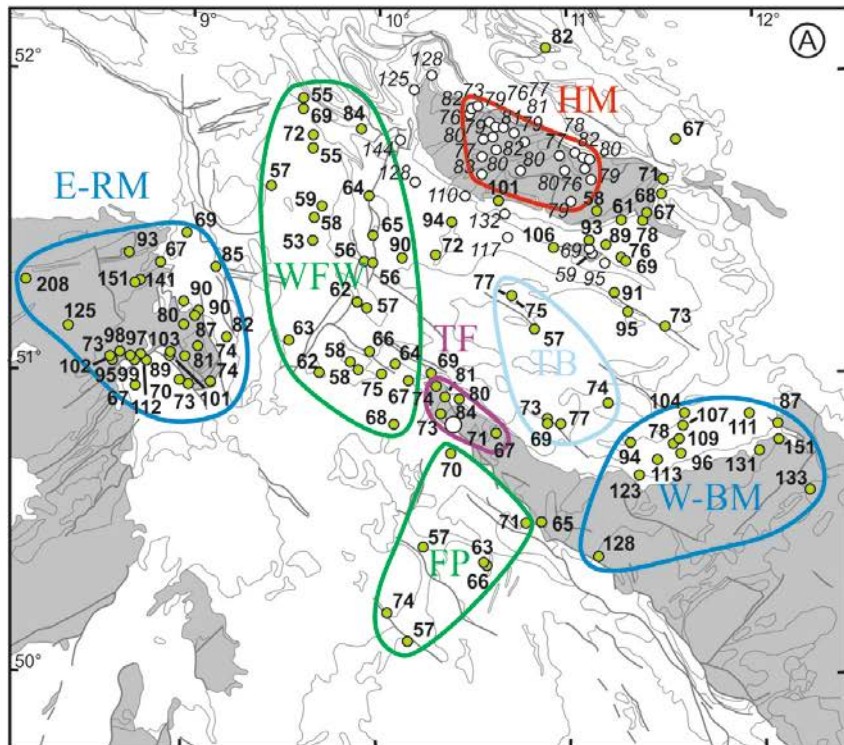

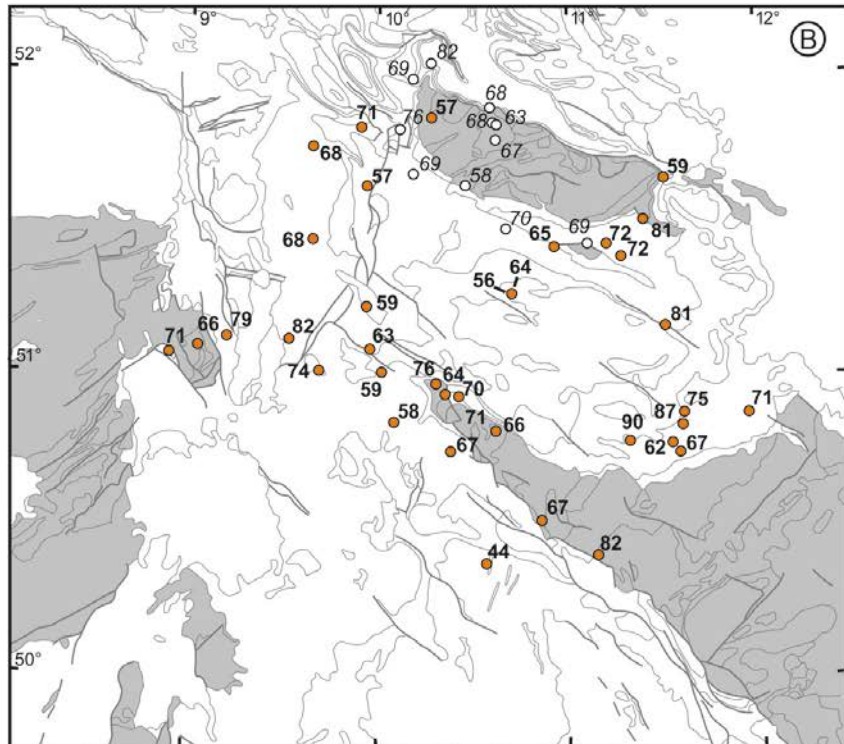

Figure 7: Spatial presentation of the new apatite fission track (A) and (U-Th)/He ages (B). The map is a simplified version of Figure 3; gray color represents Permian Rotliegend and older formations (i.e. pre-Zechstein). Colored envelopes in (A) highlight the regions defined in the text and Figure 3, where most of the samples are clustering. Small white circles in and around the Harz Mountains indicate localities of low-T thermochronology data by von Eynatten et al. (2019). The bigger white circle in the Thuringian Forest indicates the locality of the AFT samples from the Ruhla Crystalline Complex, Thomson and Zeh (2000).



The comparison of the cumulative distributions of the individual regions reveals almost identical pairwise patterns for the following regions: (i) WFW and FP, (ii) E-RM and TB + W-BM, and (iii) the thrusted basement blocks represented by the Harz and the Thuringian Forest (Fig. 8A). The age contrast between (i) and (iii) is about 10-15 Ma, rather constant over most of the age distribution. Given the high number of samples this age contrast is highly significant (Fig. 8B). At the margins of the study area (E-RM and W-BM), the apparent ages are mostly older, including high proportions of Turonian and older ages

(>90 Ma) which tend to have shorter track length (Fig. 6). However, their youngest Late Cretaceous age components, derived from the Lower Triassic bordering the Rhenish Massif and the central Thuringian Basin (TB) are similar to the youngest ages of the thrusted basement blocks (Fig. 8A).

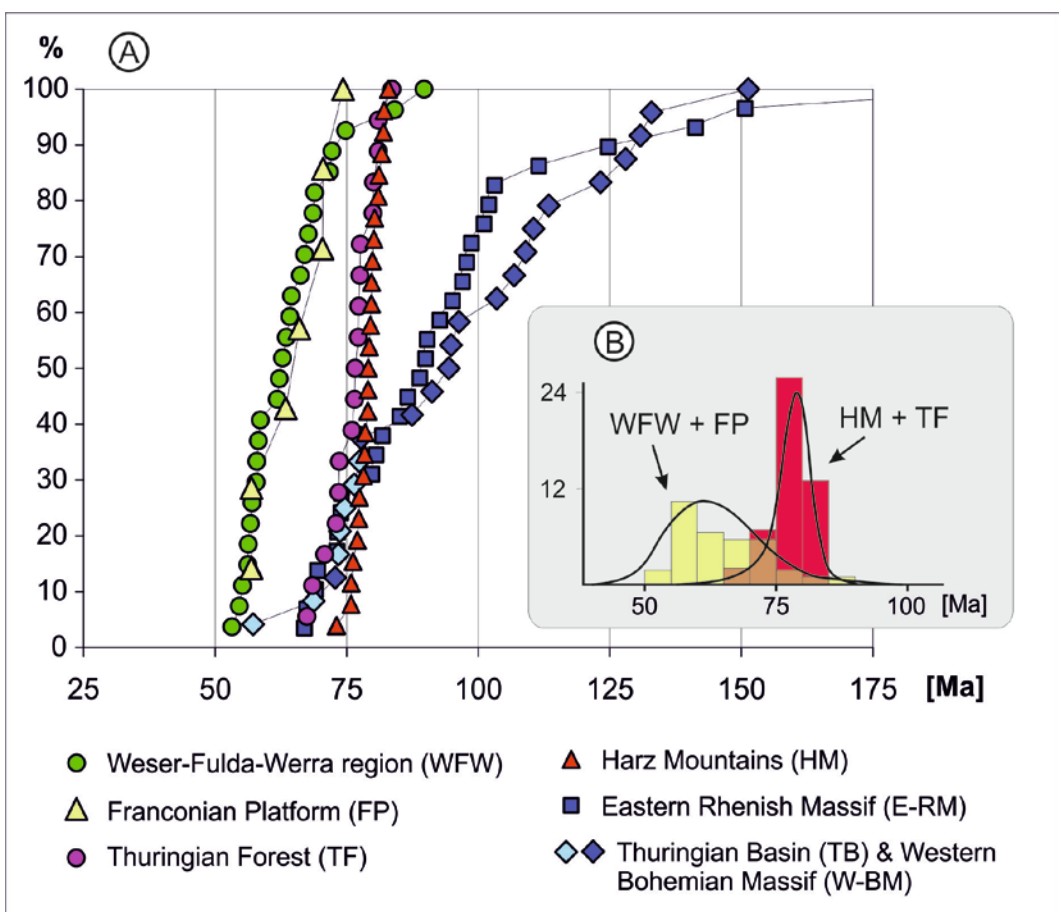

**Fig. 8: (A) Cumulative distribution of apatite fission track ages assorted for the individual regions outlined in Figure 6. Note that the central Thuringian Basin data (TB) are plotted together with its eastern transition in to the Bohemian Massif (W-BM) in a single cumulative curve, but with different color. The Harz Mountain data are taken from von Eynatten et al. (2019) and the Thuringian Forest data set is composed of our new results and data published by Thomson and Zeh (2000). (B) Histograms and kernel density estimates (KDE, Vermeesch, 2012) to emphasize the highly significant contrast of Werra-Fulda-Weser (WFW) and Franconian Platform (FP) data versus Harz Mountains (HM) and Thuringian Forest (TF) data, which differ by 15 to 20 Myr.**





### 5.3 Results of apatite (U-Th)/He analysis


The calculated unweighted average sample ages (referred to as AHe age) range from 90 to 44 Ma with the majority of ages (~62%) falling into the Late Cretaceous (Fig. 6 and Supplement S5). The rest are Paleocene ages and a single Mid-Eocene age. Standard deviations of the AHe ages vary from 1 to 12 Myr. The median of all AHe ages (67.5 Ma) is younger than the median of all AFT ages (74.7 Ma; Fig. 6). The AHe cumulative age distribution overlaps with the AFT distribution for the

youngest ages.

The spatial distribution of the AHe ages is rather uniform without significant contrast between the individual regions considered (Fig. 7B). For the core region of the study area (WFW), AHe ages range from 82 to 57 Ma, well in line with the overall range. In fact, the mean age of WFW is almost undistinguishable from the mean of all AHe ages (66.8 vs. 68.6 Ma, respectively). The two oldest AHe ages (87 and 90 Ma) occur in the eastern Thuringian Basin. However, a trend of

increasing ages towards the Thuringian Schiefergebirge (W-BM) is not observed, in contrast to the AFT data. Similarly, no trend is visible at the eastern margin of the Rhenish Massif (E-RM). The single Eocene AHe age of 44 Ma is obtained from Late Triassic sandstone of the Franconian platform (FP) with an AFT age of 63 Ma.

### 6 Thermal modelling

#### 6.1 Temperature-time paths based on low-T thermochronological data

Details of the low-T thermal history of the different regions are elucidated using the HeFTy modeling program (Ketcham, 2005), operating with the multikinetic fission track annealing model of Ketcham et al. (2007), using Dpar as a kinetic parameter. For the diffusion kinetics of helium in apatite the constraints of Farley (2000) were used. For the modelling the following constraints were considered: (i) current annual mean temperature for the surface samples and the relevant borehole temperature for the drill cores, (ii) onset of the cooling history at the late stages of the Variscan orogeny at ~300 Ma, (iii)

surface temperature of approx. 20°C at ca. 290-250 Ma for the Permian and 250-230 Ma for the Triassic sediments, and (iv) surface temperature of approx. 20°C for the onset of the Paleogene burial (ca. 45 to 35 Ma). The latter constraint is crucial for the modelling of the study area. The Paleogene sediments are widespread and seal the erosional surface of the Permo-Triassic strata especially in the west (E-RM) and east (Leipzig basin, see Fig. 5A). At least the eastern part of the Thuringian basin corresponds to a planation surface as demonstrated by scattered remnants of thin fluvial sediments of supposedly

Oligocene age ('Hochflächensedimente', Seidel, 2003, p. 415). Thus, although these areas are currently uncovered their surface can be considered as the prolongation of the Triassic/Paleogene unconformity. We allowed unsupervised run for the modeling algorithm in the time range between the Permo-Triassic and Eocene surface temperature constraints.







**Figure 9: Time-temperature (tT) plots showing the envelopes of the good results of the modeling series performed on selected samples from the different regions. The modeling has been performed by HeFTy software (Ketcham, 2005) using the AFT data. Only tT-paths classified as good fits by the program are considered. The dark gray sections represent the well constrained intervals of the cooling histories, while the light gray fields indicate the time-temperature regime where the modeling results carry insignificant information, as these ranges are older than the oldest fission tracks in the samples. Grey boxes indicate the time–temperature constraints when the dated samples experienced surface temperature conditions. The graphs in the upper right emphasize the contrast in the post-climax cooling paths between the exhumed basement blocks (A, B) and the Permo-Triassic of the WFW core region (D, E).**






The modeled temperature-time (tT) paths are rather uniform within the individual regions outlined above; Fig. 9 shows a selection covering all regions including a northern and a southern example for the core region WFW (Fig. 9D and E, respectively). A prominent feature of the tT-paths obtained in many samples is a characteristic inflexion at around 90 to 70 Ma indicating onset of cooling. It appears in all Triassic samples from the WFW, FP and TB regions, but also in the

basement samples of TF. If we zoom into the post-90 Ma cooling paths a striking difference can be recognized for the samples from WFW and Franconian Platform when compared to the samples from the Thuringian Forest and the Harz Mountains  (see pairwise comparison of the tT-paths in Fig. 9, A vs. D and B vs. E). The former show a quasi linear cooling path between the inflexion point and the age of the onlapping Paleogene sediments, while the basement highs (TF and HM) show a more hyperbolic cooling trend (i.e. initial quick cooling followed by slower cooling). This observation explains,

besides the slightly younger inflexion points, the significantly younger apparent ages of the WFW and FP samples.

At the boundaries of the study area, the eastern flank of the Rhenish Massif (E-RM) and the western flank of the Bohemian Massif (W-BM), the time-temperature paths (Fig. 9G and H) are clearly different compared to both the central part and the internal basement highs: the thermal climax is older, mostly Jurassic in age, and the Late Cretaceous inflexion is not appearing except for those samples having approx. 80 Ma or younger AFT apparent ages (Fig. 7A). The samples selected for

illustration in Fig. 9 (G and H) represent the >90 Ma AFT ages, which are dominant in the Late Paleozoic rocks detected at the margins of both massifs (W-BM and E-RM, 131 Ma and 141 Ma, respectively).

### 6.2 Reconstruction of the missing sequence

The very characteristic thermal paths described above and the lithology and thickness conditions of the post-Triassic sequences in the well-preserved basins (Fig. 5) form the base for the reconstruction of the missing sequences from the deeply

eroded regions, where only Variscan basement rocks and Permo-Triassic formations are exposed at the surface. The modeling was performed by a combination of PetroMod (Schlumberger) and HeFTy (Ketcham, 2005) software.

In the first step, the stratigraphic information from the preserved surrounding basins is crucial for the modeling. The Triassic sequences have relatively constant thickness (Figs. 4 and 5) and were thus considered as invariable for the modelling. In contrast, the Jurassic and Cretaceous strata are highly variable and thus the thickness of the late Mesozoic burial must be

considered variable for the modeling (e.g. Hoth et al., 1993). Sensitivity analyses reveal that the influence of relative proportions of the Jurassic and Cretaceous thicknesses is negligible, because their variation impacts the prograde thermal path only, while the thermal reset of the AFT age is mostly sensitive to the temperature at the deepest burial stage, i.e. around the onset of basin inversion. For constraining the lithological parameters, we used limestone for the Middle Triassic, siltstone for the Upper Triassic, shale and siltstone for the Jurassic and marl for the Cretaceous sequences. The outcome of

this step is a thermal path generated for the specific stratigraphic level, which was dated by the AFT method using the assumed burial history and heat flow.



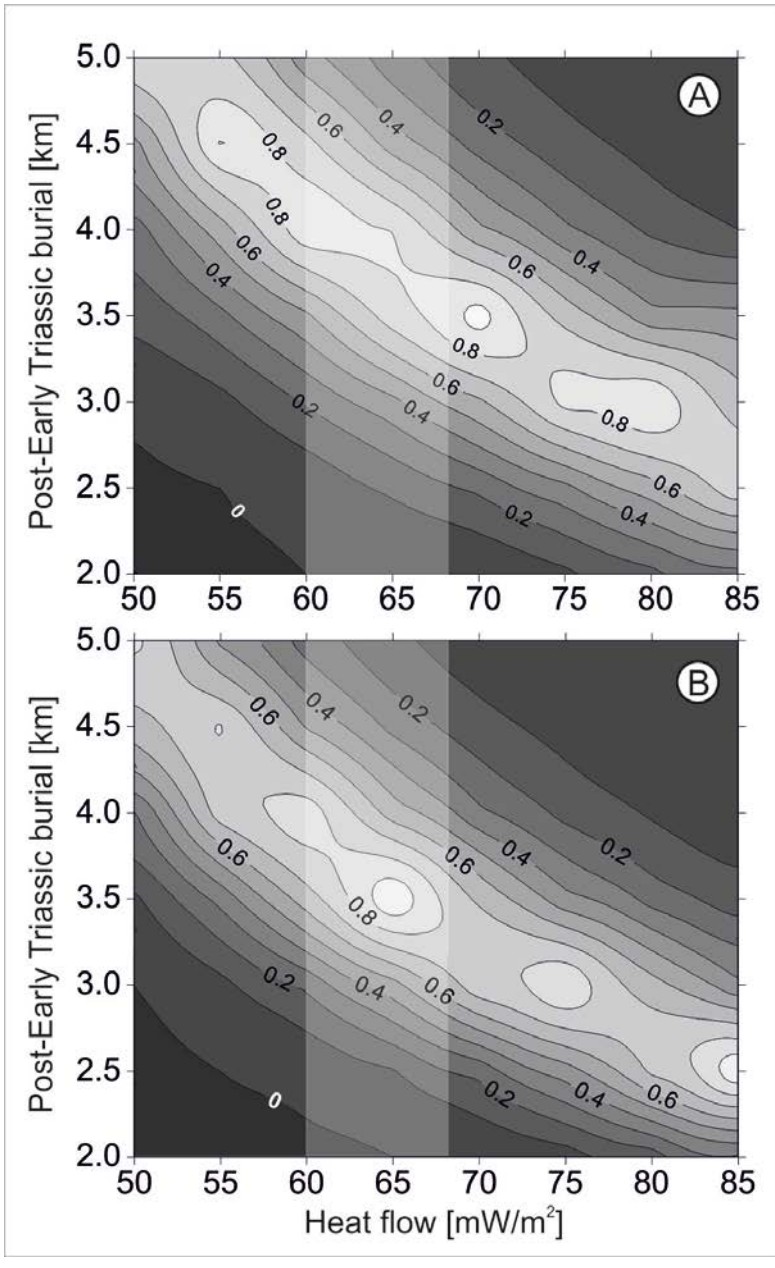

**Figure 10: Modelled relations between heat flow and the thickness of the missing sequence. The plots were generated by running 56 combinations of assumed heat flow and cover sequence thicknesses, using PetroMod (Schlumberger) and HeFTy (Ketcham, 2005) software. The isolines represent the Goodness of Fit (GoF) calculated according to the apatite fission track ages measured on the Lower Triassic Bunter sandstone samples V-27 from the Thuringian Forest (A) and V-144 from the WFW region (B). High GoF value (max = 1) indicates good match between the measured and calculated values. The burial and exhumation trends applied for the burial modeling follow the observed characteristics obtained by the thermal modeling of the Thuringian Forest and WFW samples. In case (A) the inflexion of the thermal history was set to 90 Ma followed by rapid and then decreasing cooling, while in case (B) the inflexion is set to 75 Ma and the cooling rate is kept constant until Eocene (see text and top right panels in Fig. 9). The light vertical band represents a most likely heat flow of 60 to 68 mW/m². The corresponding burial amounts to 3.5 to 4 km, actually undistinguishable for the two modeled scenarios.**

In a second step, this thermal path was used to calculate an apatite fission track age. The calculated age is then compared to
the measured age and according to the difference a misfit plot is constructed using the goodness of fit (GoF; see Fig. 10),
which expresses the probability of failing the null hypothesis that the model and data are different (Ketcham, 2005). A
detailed description of the two-step modeling procedure is given in Arató et al. (2018). Figure 10 shows the relation of the
missing burial and the heat flow calculated for a sample from the Thuringian Forest and the WFW region. The negative
correlation between the heat flow an the (paleo-)burial is obvious, the undulation and lenses in the plots are artefacts, and
related to the calculated nodes within the plot areas. Considering heat flow values of 60-68 mW/m$^2$ (see discussion below)
the removed thicknesses are in the order of 3.5 to 4 km, very similar for both the TF and WFW regions.

## 7 Discussion

In the following we first outline the principal constraints imposed by both the new and the published thermochronological
data regarding magnitudes and timing of Mesozoic to Cenozoic burial, exhumation and denudation. We then discuss the
validity of the range of assumed heat flow values in the light of Late Cretaceous to Paleogene volcanism. Eventually, we
evaluate potential driving mechanisms for the inferred uplift and exhumation based on their magnitudes and rates. Although
the discussion is focused on the main study area in central Germany, any model has to consider that the entire region which
experienced Late Cretaceous to Paleogene exhumation and erosion is considerably larger, as summarized in section 3.

### 7.1 Constraints from thermochronology and thermal modelling

Thermochronological data and modelling results in concert with sedimentological and/or stratigraphic data related to
thrusted basement blocks in Central Europe suggest exhumation and erosion of several kilometers of Late Paleozoic to
Mesozoic overburden in Late Cretaceous to Paleocene time (see section 3 and Fig. 2). For the basements blocks within or
adjacent to the main study area in central Germany the removed overburden amounts to at least 6 km in case of the Harz
Mountains (von Eynatten et al., 2019) and about 3 - 4 km in case of the Thuringian Forest. For the latter similar values are
obtained for the crystalline core (Ruhla Crystalline Complex, Thomson and Zeh, 2000) and Early Triassic sandstones from
the rim (Fig. 10A). Most of this removal has occurred in Late Cretaceous time. Similar timing and magnitude have also been
reported for the Flechtingen High to the north of the Harz Mountains (Fischer et al., 2012; Figs. 1 and 4).

The new thermochronological data presented for the Triassic sedimentary rocks exposed between and around the thrusted
basement blocks (Fig. 7) call for extensive exhumation and substantial erosion, similar in magnitude to many thrusted
basement blocks (i.e. 3-4 km assuming typical geothermal gradients and heat flow for continental crust; see Figs. 10 and
section 7.2). This regional exhumation feature holds at least for the main study area, covering about 200 x 300 km in central
Germany, referred to here as domal uplift (Fig. 11). In contrast to the basement blocks, the areas in between, reflecting the



domal uplift, are associated with significantly younger AFT ages, mostly ranging from 75 to 55 Ma (compared to 85-70 Ma for the basement blocks, Fig. 8B). This contrast is also recorded by a later onset of cooling and exhumation in the tT-paths

(Fig. 9, at ~75 Ma instead of ~90 Ma). Overall long tracks, narrow track length distributions, and mostly tight ranges of apparent ages suggest that uplift and erosion occurred over a relatively short time interval of several Myr to few tens of Myr, translating into exhumation rates of about 0.1 to 0.5 mm/yr. The 75-55 Ma age range for regional exhumation in Germany is similar to an Early Cenozoic (65-55 Ma) exhumation event affecting most of the British Isles (Holford et al., 2009a and references cited therein). Whether or not these spatially separated events reflect a common underlying process remains to be

established.

Towards the margins of the study area, at the transition to the Rhenish Massif in the west and Bohemian Massif in the east, exhumation is less pronounced and extends over a longer time span (Late Jurassic to Cretaceous). Although a general trend of increasing AFT ages towards the exposed Paleozoic basement rocks is visible (Fig. 7A), the age patterns are rather heterogeneous for these two regions, most likely because the margins of the large basement massifs are structurally complex

and the thickness and facies of the Mesozoic sedimentary coverage was variable at relatively small spatial scales. The magnitudes of Late Cretaceous to Paleocene exhumation and erosion are significantly reduced, most likely due to a combination of overall less Mesozoic burial, the lack of pronounced thrusting comparable to, e.g., the Harz Mountains, and waning uplift towards the margins of the dome. Interestingly, the tT-paths for these regions (Fig. 9G and H) are rather flat at 100 to 75 Ma implying relatively stable thermal conditions, which preclude both strong cooling due to exhumation and a

significant regional thermal pulse in Late Cretaceous time. The slight increase in temperature at around 75 to 55 Ma appears similar to the southern rim of the Harz Mountains (Fig. 9I), where it has been interpreted as temporal burial due to storage of detritus delivered from the exhuming Harz Mountains (von Eynatten et al., 2019).

## 7.2 Volcanism and heat flow

Mostly Tertiary alkaline basic volcanic rocks such as basanites, nephelinites and alkali olivine basalts, along with minor

occurrences of more differentiated rocks such as phonolites and trachytes characterize the Central European Volcanic Province (CEVP; Wilson and Downes, 1991; Wedepohl et al., 1994). It includes major occurrences of volcanic rocks like the Eifel, Westerwald, Vogelsberg, and Rhön in central Germany (Fig. 3) and the Eger rift in the Czech Republic. Published ages vary from Middle Eocene to Quaternary, however, most occurrences are Oligocene–Miocene in age (e.g. Wedepohl et al., 1994). The Miocene Vogelsberg volcano in northern Hesse, dated between 18 and 14 Ma (Bogaard and Wörner, 2003),

forms the center of the CEVP with respect to both location and volume. The geochemical and isotopic characteristics of the volcanic rocks mainly suggest derivation from an upwelling asthenospheric mantle source, although some part of the melts indicate a cooler origin in the lithosphere affected by asthenospheric melts ('veined lithospheric mantle'; Boogaard and Wörner, 2003; Jung et al., 2005). The more differentiated rocks of the CEVP suggest variable crustal contamination (e.g. Kolb et al., 2012; Jung et al., 2013).






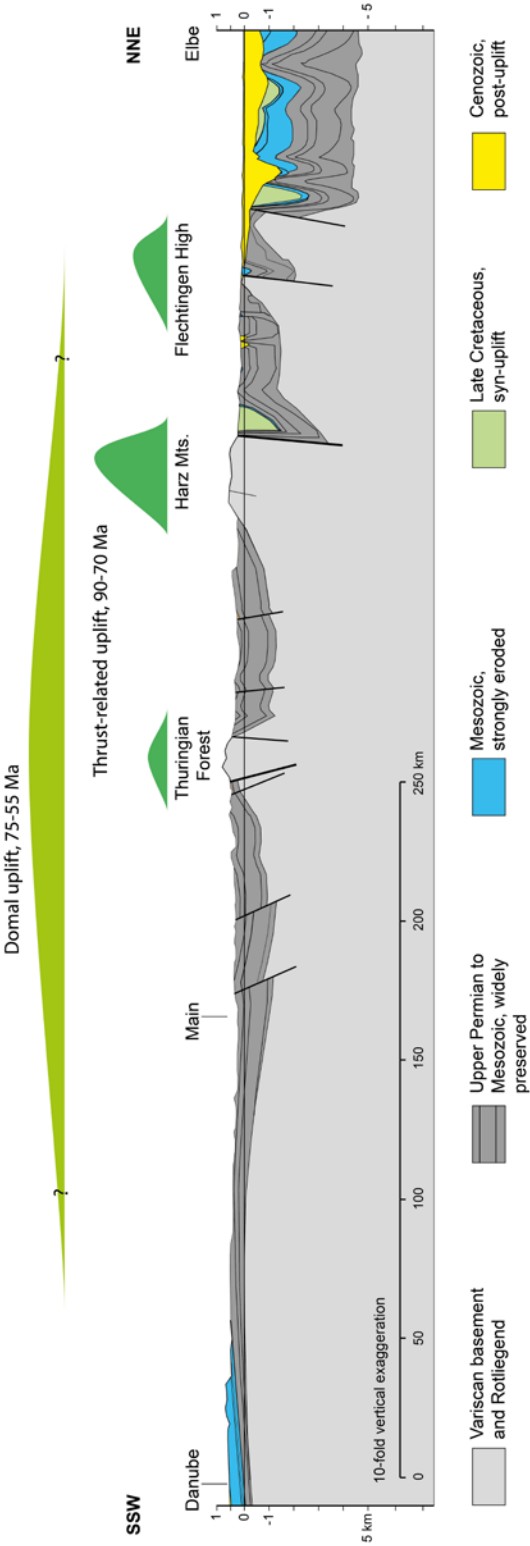

**Figure 11: The two superposed uplift and exhumation processes illustrated on the cross-section of Fig. 4. Thrust-related uplift confined to three discrete basement blocks is followed, possibly with some temporal overlap, by domal uplift persisting into the Paleogene. Uplift and exhumation magnitudes are not to scale.**



Although the CEVP is mainly a Late Eocene to Miocene feature, there is widespread evidence for Late Cretaceous to Paleocene volcanic activity in the entire area. This includes zircon U-Pb ages of 70 to 67 Ma from trachytes and syenites from the Vogelsberg area and further occurrences to the north and south of the Odenwald (Schmitt et al., 2007; Martha et al.,

2014). Similar ages (68 and 69 Ma) are reported from Ar/Ar dating of amphibole from camptonite dikes cutting the Paleozoic-Mesozoic basement of the Vogelsberg volcano (Bogaard and Wörner, 2003). Further south, along the margins of the Upper Rhine Graben, Ar/Ar dating of basic alkaline volcanic intrusions and dikes revealed Paleocene to Early Eocene ages between 61 and 47 Ma (see compilation in Walter et al., 2018). Siebel et al. (2009) reported radiometric data from zircon crystals derived from alkaline basalts of the Eger Rift. Whereas the (U-Th)/He cooling ages of these zircons reflect

the well-known Late Oligocene eruption of the lavas, their U-Pb data suggest crystallization ages ranging from 83 to 51 Ma. The data reflect multiple or protracted zircon growth events or reset at different times long before eruption and call for survival of the in zircons in a locally enriched chemical environment affected by asthenospheric upwelling processes or zircon growth in a metasomatically enriched subcontinental lithospheric mantle (Siebel et al., 2009).

The contrast between the inferred eroded thicknesses from thermochronological data of the Thuringian Forest and the

significantly lower thicknesses derived from stratigraphic data of the adjacent basins led already Thomson and Zeh (2000) to speculate about (i) hitherto unreported thicknesses of Jurassic to Lower Cretaceous strata (which in required thickness are only available in the Lower Saxony Basin and small confined basins like the Harz Foreland Basin, Fig. 5), and/or (ii) significantly increased geothermal gradients. The latter, however, would have to be as high as ~55-60°C/km to fully account for the temperatures attained under a thin overburden. Assuming common thermal conductivity values for upper crustal

rocks of 2-3 W/mK (e.g. Mielke et al., 2017), the corresponding heat flow would be about 110 to 180 mW/m$^2$, even higher than the extreme values recorded in the Pannonian Basin or most of the Basin-and-Range Province at present (Lenkey et al., 2002; Sass et al., 1994). Northeast of the study area, in the NEGB, present heat flow values range about 70 to 90 mW/m$^2$ (Norden et al., 2008). Towards the study area, these values appear to decrease to thermal gradients around 30°C/km (Agemar et al., 2012, their figures 8 and 11). However, data coverage regarding present subsurface temperature distribution is rather

poor for the central German uplands region. Thermometric and thermochronological studies from the Rhenish Massif call for a 'normal' and stable geothermal gradient since late Paleozoic times (Glasmacher et al., 1998; Karg et al., 2005). Further north, in the strongly inverted central and southern part of the LSB, Senglaub et al., (2005) assumed a slightly elevated gradient of about 40°C/km during the time of maximum burial (i.e. Jurassic to Early Cretaceous).

To summarize, there is clear evidence that Central Europe was affected by alkaline intraplate volcanic activity in latest

Cretaceous to Early Eocene time. The volumes are negligible with respect to heating of the crust implying that our thermochronological data are not directly influenced by volcanic heat production. Moreover, there is no other clear evidence for large-scale and long-term increased or reduced heat flow in central Europe. Therefore it seems justified to assume a typical gradient of ~30°C/km and heat flow of ~65 mW/m$^2$ for the modelling. The widespread witnesses of volcanic activity,



although thermally insignificant, point to processes in the asthenospheric mantle and/or across the asthenospheric-
lithospheric mantle boundary that likely affected geodynamics and uplift in Central Europe at that time.

**7.3 Potential mechanisms for uplift, exhumation and denudation**

The observables to be explained by a model for exhumation away from the thrusted basement uplifts are domal uplift of an
area at least some 250-300 km across, with concomitant denudation of 3-4 km. Uplift and erosion occurred over a relatively
short time interval of several Myr to few tens of Myr, corresponding to exhumation rates of 0.5 to 0.1 mm/yr. Afterwards,
the crust subsided slightly as indicated by the remnants of Cenozoic marine strata, but remained significantly uplifted above
its previous elevation and has not accommodated a new, thick sedimentary cover to the Present. With respect to timing, this
event occurred approx. 10 to 20 Myrs after thrusting of the basement blocks (Figs. 8 and 9).

In general, mechanisms capable of inducing long-wavelength uplift of the continental crust can be grouped into two
categories: (1) Isostatic. Here, the elevation change is caused by variations in the density and/or thickness of lithospheric
materials or the asthenosphere. The crust can be thickened via tectonic shortening or addition of melt (Brodie and White,
1994; Ware and Tuner, 2002). Isostatic mechanisms involving the mantle comprise thinning of the lithospheric mantle by
heating (thermal erosion) and translation of the lithosphere over more buoyant asthenosphere (Carminati et al., 2009). (2)
Dynamic. Uplift in this case is due to viscous stress from upwelling mantle or results from buckling of the crust or entire
lithosphere under tangential tectonic stress.

For most of the mechanisms listed above it is relatively straightforward to estimate whether they could have created uplift of
sufficient magnitude and rapidity. We conservatively consider 4 km of uplifted crust that are fully eroded (no topography).
In the case of wholesale lithospheric uplift, the base of the lithosphere is raised by h = 4 km, creating room for 4 km of
asthenosphere at the bottom of the column while an equal amount of crust is eroded from the top (Fig. 12A). The pressure
increase equals the load of the added asthenosphere minus the load of the eroded crust following Eq. (1):

$$\Delta p = \rho_{ast}*g*h - \rho_c*g*h \quad \text{or } \Delta p = (\rho_{ast} - \rho_c)*g*h \tag{1}$$

where $\rho_{ast}$ and $\rho_c$ are the densities of the asthenosphere and continental crust and g is gravity acceleration. The pressure
increase $\Delta p$ must be compensated by an equal but upward directed force per area which can be created by addition of less
dense material (for instance, basalt instead of mantle rock) or removal of denser material (for instance, mantle lithosphere
replaced by asthenosphere). As $\Delta p$ is the product of a density difference and thickness (and constant gravity acceleration),
the required thickness change for any material added or lost can be calculated using Eq. (2):

$$(\rho_{ast} - \rho_c)*g*h = (\rho_{ast} - \rho_x)*g*x \quad \text{or, eliminating g and rearranging,}$$

$$x = h *(\rho_{ast} - \rho_c) / (\rho_{ast} - \rho_x) \tag{2}$$

where x and $\rho_x$ are the thickness and density of the material added or removed. The results are listed in Table 1 and briefly
discussed in the following. For the sake of simplicity we have chosen single density values rather than ranges. The results do
not change fundamentally if densities are varied within reasonable limits.





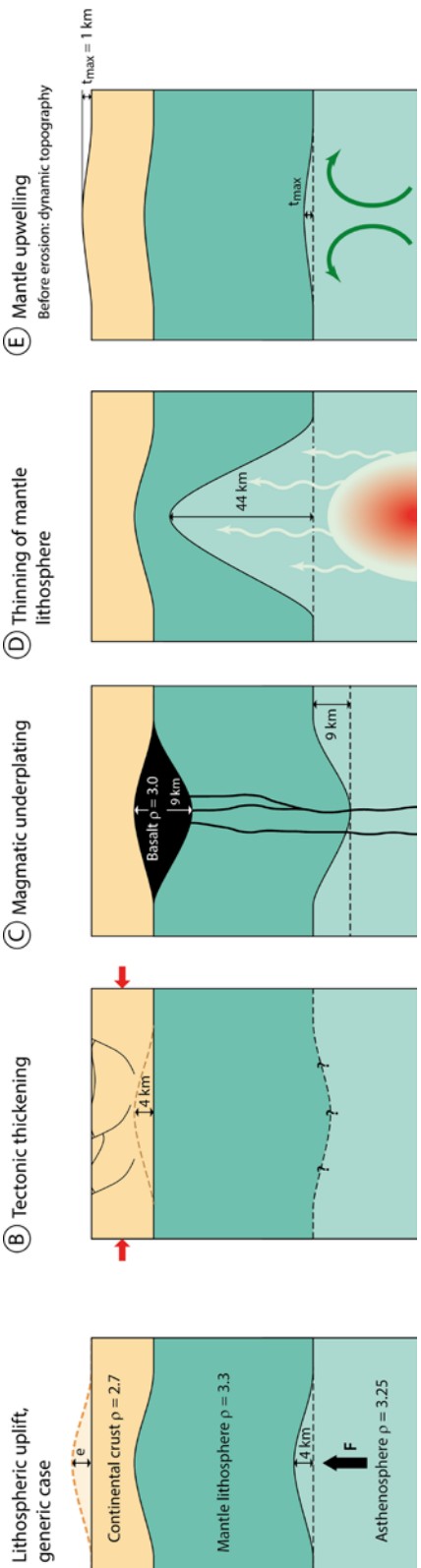

**Figure 12: Scenarios for domal uplift inducing 4 km of erosion as discussed in the text. All indicated density values are in g/cm³. Sketches are not to scale. (A) Generic case of lithospheric uplift. 4 km of crust are eroded and 4 km of asthenosphere added; thickness of lithospheric mantle is unchanged. (B) Crust thickened by folding and thrusting restored to original thickness by erosion. Tectonic thickening of lithospheric mantle depends on its original thickness and was not considered in our estimate. (C) Underplating by basaltic melt. Thickness of lithospheric mantle unchanged. (D) Thinning of mantle lithosphere by thermal erosion. (E) Dynamic topography caused by mantle upwelling. After erosion, the configuration is similar to (A), with asthenosphere replacing continental crust. In nature, (C), (D) and (E) are not mutually independent. See Table 1 for derivation of numerical values.**





**Table 1: Estimates of parameters for mechanisms of uplift and exhumation discussed in the text.**

| Mechanism | Material x | Density (g/cm3) | replacing | Density (g/cm3) | Density difference past − px | Density difference ratio (past − pc) / (past − px) | Thickness change*(km) | Pressure change (MPa) |
|---|---|---|---|---|---|---|---|---|
| Tectonic thickening | Continental crust | 2,7 | Asthenosphere | 3,25 | 0,55 | 1 | 4 | 22 |
| Magmatic underplating | Basalt | 3 | Asthenosphere | 3,25 | 0,25 | 2,2 | 8,8 | 22 |
| Lithospheric thinning | Asthenosphere | 3,25 | Lithospheric mantle | 3,3 | -0,05 | -11 | -44 | 22 |
| Upwelling mantle: | | | | | | | | |
| Surface uplift (no erosion) | Asthenosphere | 3,25 | Air | 0 | 3,25 | 0,17 | 1 | 32,5 |
| Erosion of crust | Asthenosphere | 3,25 | Continental crust | 2,7 | 0,55 | 1,00 | 5,9 | 32,5 |

* Thickness change calculated for 4 km of exhumation, except upwelling mantle case where it is calculated from 1 km of dynamic topography





### 7.3.1 Tectonic thickening of the crust

Thrusting and folding would have to thicken the crust by 4 km before erosion restores it to its original thickness (Fig. 12B). Increasing the crustal thickness in the core region from its present (and, by inference, also pre-inversion) value of 30 km to 34 km requires shortening by 12%. This is a conservative estimate because we assume no topography is left at the end of erosion and do not consider the subsidence caused by tectonically thickened lithospheric mantle. The total shortening accommodated by folding and thrusting in Germany during Late Cretaceous inversion has been estimated at 13.5 to 15 km (Jähne et al., 2009), corresponding to an average percent shortening across the 250 km-wide uplift of around 6 %. Crustal shortening could thus make a significant contribution to exhumation but not explain its full amplitude, even if the shortening accommodated by individual inversion structures may be underestimated (Eisenstadt and Withjack, 1995; Holford et al., 2009b, cf. Bolz and Kley, this volume). The need for an additional uplift mechanism is also evidenced by the observation that structural lows such as the Thuringian Basin (syncline) did not subside but were strongly exhumed (Fig. 4). As regards the uplift rate, tectonic shortening thickens the crust at the shortening rate divided by the aspect ratio of the deforming cross-section area, in our case about 10 (300 km length/30 km thickness). 1 mm/yr of shortening would give 0.1 mm/yr of thickening and it would take 40 Myr to thicken the crust by 4 km. Notice, however, that this shortening rate derives from the estimated magnitude of less than 20 km shortening and does not provide an independent constraint on the capability of crustal thickening to drive exhumation.

### 7.3.2 Magmatic underplating

Isostatic uplift due to underplating by basaltic melts has been proposed for parts of the British Isles (e.g. Brodie and White, 1994; there associated with the Iceland plume). The ca. 9 km thick column of basalt required to drive 4 km of upper crustal erosion (Fig. 12C, Table 1) is similar to the thickness of layered mafic lower crust observed in many parts of Central Europe. However, this lower crust is commonly interpreted to reflect crustal re-equilibration in Permian time (e.g. Lüschen et al., 1990) and it seems unlikely that the crust before Cretaceous uplift was only 25 km thick (calculated as the pre-erosion thickness of 34 km minus 9 km of basalt). The attainable uplift rate depends on the rate of magma production. Underplating the entire area of Cretaceous uplift (ca. 70.000 km$^2$) by several km of basaltic melt would require a magma volume typical of Large Igneous Provinces (LIP; e.g. Stein et al., 2018, their Fig. 8). As the production of basaltic melts can be very fast and continental flood basalts are typically erupted over only a few million years (Self et al., 2014), the emplacement rate is not limiting. However, the sparse Late Cretaceous to Paleocene volcanism makes Central Europe a highly unlikely location for a LIP of that age.

### 7.3.3 Thinning of the lithospheric mantle

The lithospheric mantle must thin by several tens of kilometers to induce 4 km of erosion due to the small density contrast of lithosphere and asthenosphere (Fig. 12D, Table 1). Thickness variations of the lithospheric mantle over time have been





invoked to explain Mesozoic subsidence and Cenozoic uplift in Central Europe (Meier et al., 2016), but the magnitude of
these changes is difficult to constrain independently. Thinning of the lithospheric mantle by upwelling asthenosphere
strongly depends on the size of the uplifted area and thus the distance over which the asthenosphere spreads laterally. Semi-
analytical models (Davies, 1994) suggest that an initially 120 km thick continental lithosphere can be thinned to less than 60
km within 25 Ma above a narrow plume of 70 km diameter, creating 1.5 km of non-eroded uplift, but thinning and uplift are
much less efficient for an area 300 km across. However, Davies (1994) also pointed out that thinning of the lithosphere by a
thermal plume is preceded by faster dynamic uplift as the plume reaches the base of the lithosphere (see also Friedrich et al.
(2018) for predicted geological effects of mantle plumes). Extremely fast removal of the continental lithosphere over only a
few hundred thousand years by an impinging thermochemical plume that contains a large amount of recycled oceanic crust
was modeled by Sobolev et al. (2011) but creates only about 200 m of non-eroded uplift.

### 7.3.4 Relative motion of lithosphere and asthenosphere

Westward absolute motion of the European plate over asthenosphere of lower density created at the Mid Atlantic Ridge was
proposed as a cause of Cenozoic uplift (Carminati et al., 2009). In Late Cretaceous time the MOR of the North Atlantic did
not exist yet, and central Europe cannot have moved over depleted asthenosphere formed there. However, absolute motion
could have moved central Europe over long-lived dynamic topography (see next paragraph).

### 7.3.5 Dynamic topography

Isostatic calculations can also be used to estimate uplift and exhumation due to mantle flow. Observational data suggest that
upwelling mantle is able to sustain about 1 km of dynamic topography t (Braun et al., 2013), or 1 km of asthenosphere added
to the base of the lithospheric column. If the topography is eroded while the mantle flow is maintained, the eliminated crustal
load allows for additional uplift of mantle. Eventually, the load of the mantle replacing eroded crust equals that of the mantle
column before erosion following Eq. 3 (Fig. 12A, 12E):

$$e * (\rho_{ast} - \rho_c) = \rho_{ast} * t \quad \text{or} \quad e = t * \rho_{ast} / \rho_{ast} - \rho_c \tag{3}$$

where e is eroded crustal thickness, t is dynamic topography and $\rho_c$, $\rho_{ast}$ are the densities of crust and asthenosphere.
Mantle upwelling capable of sustaining 1 km of dynamic topography could thus drive some 6 km of erosion (Table 1).
The rates at which dynamic topography grows and decays depend on the mechanism assumed: Topography can result from
(i) a lithospheric plate moving across a stable upwelling area or swell related to long-term mantle convection (Braun et al.
2013) or (ii) a plume-like perturbation rising beneath the lithosphere. In the first case, the uplift rate depends on absolute
plate velocity and the geometry of the swell. In the second case it depends on properties of the plume. The absolute motion
of Eurasia underwent major changes between 100 and 60 Ma, from a fast northwest to a slower southwest and eventually
slow west-directed motion, at maximum velocities of about 5 mm/yr for central Europe (Seton et al., 2012). At 5 mm/yr it
would take 60 Myr to move a 300 km wide area onto a swell.





### 7.3.6 Lithospheric folding

The potential contribution of lithospheric folding to domal uplift is more difficult to assess because its amplitude strongly
depends on lithosphere age and rheology as well as the magnitude and rate of shortening. Numerical modelling results
indicate that folding-induced erosion can attain magnitudes way beyond the 4 km discussed here with sufficient shortening
and time. On the other hand, a generic model approximately matching our case of a 300 Ma old lithosphere develops non-
eroded topography of only 200 m amplitude at 6% of shortening (a 1500 km wide model area shortened by 90 km; Cloetingh
and Burov, 2011, their Fig. 10), despite a shortening rate of 15 mm/yr, more than ten to a hundred times higher than during
Late Cretaceous time in Germany. At low shortening rates lithospheric folds take time to grow (e.g. 5-10 Myr for 1 km
amplitude at 4 mm/yr for Iberia, Cloetingh et al., 2002). The strongest argument against a major role of lithospheric folding
in our case, however, is that a region that was subsiding until the onset of inversion will not become uplifted but exhibit
accelerated subsidence under tangential compression.

### 8 Conclusions

- A compilation of several hundred published and about 150 new thermochronological analyses (AFT and AHe) indicates
generalized, km-scale exhumation over substantial parts of Central Europe in Late Cretaceous to Paleocene time.
- The magnitude of exhumation attains >6 km over basement uplifts such as the Harz Mts. and 3-4-km in the other regions in
central Germany.
- The spatial pattern of exhumation exhibits two types of exhumation: (i) thrust-bordered basement uplifts and (ii)
superimposed regional-scale domal uplift. While thrust-related exhumation is spatially well defined, the extent of the doming
area is as yet poorly defined.
- In detail, thrust-related uplift predates regional doming (90-70 Ma vs. 75-55 Ma) with some temporal overlap.
- Thrusting and crustal thickening associated with inversion tectonics can contribute at best half of the doming signal. An
additional process or processes are required to explain the widespread exhumation.
- Thinning of the mantle lithosphere and dynamic topography, both caused by upwelling asthenosphere, are able to produce
uplift of the required magnitude, wavelength and rate. The exhumed region does coincide with a raised lithosphere-
asthenosphere boundary at present.
- Alkaline volcanism potentially associated with mantle-induced uplift is dated at about 70 to 50 Ma, roughly similar to
domal uplift, but it's scarcity and negligible volume is puzzling.
- The apparent southern border of the Southern Permian Basin is due to exhumation. Its original depositional realm extended
much further to the south.
Additional data are needed to constrain the spatial extent of domal uplift and exhumation and to decide whether it is an
isolated occurrence or linked to other regions of Paleogene uplift such as the British Isles or the western and northeastern
margins of the Bohemian Massif. The temporal contrast between the well-known Late Cretaceous thrusting in Central



Europe and the newly discovered, younger, long wavelength domal uplift should also be investigated, verified or falsified, for other regions in Central Europe.

*Supplement*. The supplement related to this article, including sample data and thermochronological data (Figures and Tables
S1 to S5), is available online at: …..........

*Author contributions*. HvE and JK designed the study. VH and ID performed the fission track analyses, AS and ID performed the (U-Th)/He analyses. ID performed the thermal modelling. HvE and ID evaluated and interpreted the thermochronological data, JK compiled and interpreted structural data. HvE, JK and ID jointly prepared the manuscript.

*Competing interests*. The authors declare that they have no conflict of interest.

### Acknowledgements

The study has been partially funded by the German Research Foundation (grants EY23/9 and KL495/9). We appreciate great support by Irina Ottenbacher and Judit Dunklné-Nagy during sample preparation and mineral selection. Thomas Voigt (Jena)
helped us during fieldwork and sampling. Fabian Jähne-Klingberg (BGR, Hannover) contributed to the structural geology work in the early phase of this project and helped us to develop our ideas in many later discussions.

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

**S3, Table: Apatite fission track data (n=110)**

**S4, Figure: Binned diagrams of horizontal confined fission track lengths measured in the apatite samples.**

**S5, Table: Apatite (U-Th)/He data (n=37).**