# Peer review of "Late Cretaceous to Paleogene exhumation in Central Europe – localized inversion vs. large-scale domal uplift"

_Solid Earth, 2020_

## Referee Comment (RC1) · Christoph von Hagke (Referee) · 11 Dec 2020

This manuscript presents a compilation of thermochronological data from Central Europe, strengthened by roughly 150 new AFT & AHe data. Using this data, the uplift and exhumation history of Central Europe is constrained, and different driving mechanisms are discussed based on thermal modeling and rough calculations of the response signals to isostatic and dynamic processes. The study concludes that a combination of thrust related exhumation and large-scale domal uplift explain best the data.

The study presents an effort that is comprehensive, of timely importance and is very well written. It should be published after some very minor corrections.

[Figure]

Figure 1: Does not work well in b&w. The profile is very schematic, and more detail should be added; The fault at the northern fringe of the TF is not shown in the map, or it should be located in the U-Permian section.

Figure 2: This is an interesting plot, but some revisions would be good to make it more accessible. Currently on y-axis you plot number of samples. Instead you should use % (as you do in Fig 8). Error bars on ages are missing. Alternatively, you could simply use the fromat of Fig. 8B, which is very straight forward to read. I am skeptical about the meaning of median ages. For calculating the median you pool ages that are unrelated. While it does make more sense for very steep curves, a median age e.g. for the Erzgebirge seems geologically meaningless.

Thermal modeling: The hyperbolic cooling trend is visible in TF, but not so much in HM. I find it unfortunate that you present envelopes only, as the single path plot would show this better.

Reconstruction of missing sequence: in line 495 ff you discuss that thickness of the Jurassic to L-Cretaceous strata was possibly thicker. How would this influence your thermal model, as temperature at deepest burial would increase?

Dynamic topgraphy: You discuss plate movements of Eurasia citing Seton et al. 2012. This is a great paper, however a global model, which often cannot take into account more local results. Aren't there more local studies constraining plate movements for that particular region (ideally also in a global reference frame)?

The text is full of abbreviations. I suggest to get rid of most of them. Often not needed, and makes the text harder to follow.

There is mixed used of AE & BE (gray v grey; modeling v modelling...)

Very minor comments: Line 27: add references

Line 46: this must have been said also earlier than 1997

[Figure]

Line 97: add that few samples are from drill holes or specify near-surface to <500 m.

Line 113: the right side, not the left side. You could also say the eastern side (not sure right and left even though used in Germany is suitable here. Maybe it is....)

Line 139: " by numerous studies (as reviewed below)"

Line 168: here and elsewhere - I find the word significant overused and pushy. suggest to not use it but be quantitative instead.

Line 180: reference missing

Line 239: Reference missing

Caption Fig. 4 and elsewhere - are page numbers required after the ref.?

Line 594: put "t" in italcis

Line 608 ff: you might consider including the reference of Bourgois et al., maybe particularly as you disagree with this interpretation, and it is a well-known paper.

––––––––––––––––––––––––––––

---

## Referee Comment (RC2) · Anonymous Referee #2 · 15 Dec 2020

I believe that this manuscript is very timely in view of current efforts in understanding large-scale exhumation of large continental areas, particularly I the light of current discussions on dynamic topography effects. I appreciate the solid-written and argumented character of the manuscript, the documentation by detailed and state of the art thermochronology and the nice discussion on genetic mechanisms. I suggest that the manuscript can be accepted almost as is. What can be improved is a better link between the various genetic mechanisms discussed and a preferred solution. The validity of some of these mechanisms is not really fully clear in the manuscript. For instance, I would see lithospheric folding as fairly suitable mechanism providing an advanced explanation. However, the authors discard this mechanism because "a region

that was subsiding until the onset of inversion will not become uplifted but exhibit ac-
celerated subsidence under tangential compression", which is an unclear argument.
This is either not well explained or incorrect: sure that subsidence may be enhanced
by lithospheric folding in basins, we see such effects in many worldwide places. In a
similar way, other potential mechanisms are not fully clear in the manuscript, at least to
me. Therefore, to increase the impact of the paper, I suggest to revise, explain better
and be more quantitative to all mechanisms explained in Section 7. Otherwise, as said
above, this is a very nice contribution that fits perfectly the scope of the journal.

---

## Author Comment (AC1) · 16 Feb 2021

Response to Reviewer-1: Christoph von Hagke:

- This manuscript presents a compilation of thermochronological data from Central Europe, strengthened by roughly 150 new AFT & AHe data. Using this data, the uplift and exhumation history of Central Europe is constrained, and different driving mechanisms are discussed based on thermal modeling and rough calculations of the response signals to isostatic and dynamic processes. The study concludes that a combination of thrust related exhumation and large-scale domal uplift explain best the data. The study presents an effort that is comprehensive, of timely importance and is very well written.

[Figure]

It should be published after some very minor corrections.

Thanks a lot for the careful and positive evaluation.

- Figure 1: Does not work well in b&w. The profile is very schematic, and more detail should be added; The fault at the northern fringe of the TF is not shown in the map, or it should be located in the U-Permian section.

We have changed the colors now showing only the brown colored 'Variscan basement and Lower Permian' in the lower map, Fig. 1C. The section in Fig 1B is a simplified sketch to illustrate the overall structural pattern. The northern margin of the TF is indeed rather complex, strongly variable along strike. We now refer in the caption to the more detailed section in Fig.4.

- Figure 2: This is an interesting plot, but some revisions would be good to make it more accessible. Currently on y-axis you plot number of samples. Instead you should use % (as you do in Fig 8). Error bars on ages are missing. Alternatively, you could simply use the fromat of Fig. 8B, which is very straight forward to read. I am skeptical about the meaning of median ages. For calculating the median you pool ages that are unrelated. While it does make more sense for very steep curves, a median age e.g. for the Erzgebirge seems geologically meaningless.

The y-axis has been changed to percentage as requested. These are overview plots compiled from in part pretty old literature data, where errors are treated differently. Adding these errors would not help much given the purpose of the figure, i.e. reviewing evidence for Late Cretaceous cooling all over Central Europe.

- Thermal modeling: The hyperbolic cooling trend is visible in TF, but not so much in HM. I find it unfortunate that you present envelopes only, as the single path plot would show this better.

We prefer to keep the envelopes because presentation of the individual t-T lines may be rather misleading. Even accepted time-temperature trials may have unrealistic zig-
zag character. Such sharp turns from cooling to heating are not a reliable scenarios for the thermal evolution of sedimentary basins where the isotherms are typically moving rather gently through time. The modelling procedures offer an operator-determined limitation for the heating-cooling rates, but actually we have not any acceptable reasoning for applying maximum values, especially not, as it can modify the final thermal path. Our procedure relies on simply chopping off the meaningless sharp peaks and turns of mathematically correct but geologically unrealistic trials. In this way we emphasize the envelop of the highest density of acceptable or good thermal paths as a kind of smoothing procedure that keeps the essential character of the t-T array, but does not show the unrealistic solutions.

- Reconstruction of missing sequence: in line 495 ff you discuss that thickness of the Jurassic to L-Cretaceous strata was possibly thicker. How would this influence your thermal model, as temperature at deepest burial would increase?

Indeed, the inferred removal of 3-4 km overburden requires relatively thick Mesozoic strata including large contribution from Jurassic and Lower Cretaceous strata. For the model, their variation would impact the prograde thermal path only, which in any case should have reached AFT reset before onset of inversion/cooling in Late Cretaceous time (except for the marginal regions in the West and East). The impact of having more burial than necessary for full reset is negligible. This point is explained in section 6.2.

- Dynamic topography: You discuss plate movements of Eurasia citing Seton et al. 2012. This is a great paper, however a global model, which often cannot take into account more local results. Aren't there more local studies constraining plate movements for that particular region (ideally also in a global reference frame)?

We are not aware of studies deducing absolute plate motions based on a scale smaller than global. For instance, discrepancies between the Atlantic-Indian and Pacific hotspot reference frames need to be resolved globally.

- The text is full of abbreviations. I suggest to get rid of most of them. Often not needed,

and makes the text harder to follow.

We have omitted some abbreviations (NEGB, CEVP, ...), but decided to keep those which are frequently used by many scientists (AFT, AHe, tT-path, etc.) and those which refer to the specific sub-regions of the study area because these are used consistently throughout the text, figures and tables.

- There is mixed used of AE & BE (gray v grey; modeling v modelling...)

Corrected to BE

- Very minor comments: Line 27: add references

Done

- Line 46: this must have been said also earlier than 1997

Yes, that's true. We now also cite Ziegler 1987 (and references therein).

- Line 97: add that few samples are from drill holes or specify near-surface to <500 m.

Done

- Line 113: the right side, not the left side. You could also say the eastern side (not sure right and left even though used in Germany is suitable here. Maybe it is....)

Corrected to right side

- Line 139: " by numerous studies (as reviewed below)"

Done

- Line 168: here and elsewhere - I find the word significant overused and pushy. Suggest to not use it but be quantitative instead.

Ok, the use of the word significant is now strongly reduced, replaced by words like marked, distinct, remarkably, etc.

[Figure]

- Line 180: reference missing

It's the same references as for the sentence before; Vamvaka et al. 2014, now added.

- Line 239: Reference missing

We now refer to Lotze (1948) and Kley et al. (2008).

- Caption Fig. 4 and elsewhere - are page numbers required after the ref.?

Not sure. I guess they are not required but usually help the readers in case of long papers, chapters or textbooks. We decided to keep them and leave the final decision for the editorial handling. . .

- Line 594: put "t" in italics.

Done

- Line 608 ff: you might consider including the reference of Bourgois et al., maybe particularly as you disagree with this interpretation, and it is a well-known paper.

Bourgeois et al. (2007, in IJES) discuss what they interpret as present-day lithospheric folds trending SW-NE, caused by the present stress field acting since ca. 25 Ma. We would like to avoid entering a discussion on whether and how hypothetical Cretaceous lithospheric folds would give way to folds of a new direction, how long that would take (Burov and Cloetingh in one of their papers discuss the persistence of lithospheric folds), and whether traces of Late Cretaceous to Paleocene folds could be preserved. We feel that would take us too far away from our line of reasoning. Anyway, there is a more detailed discussion of lithospheric folding in the new version (see also response to reviewer-2).

---

## Author Comment (AC2) · 16 Feb 2021

Response to Reviewer-2:

- I believe that this manuscript is very timely in view of current efforts in understanding large-scale exhumation of large continental areas, particularly I the light of current discussions on dynamic topography effects. I appreciate the solid-written and argumented character of the manuscript, the documentation by detailed and state of the art thermochronology and the nice discussion on genetic mechanisms. I suggest that the manuscript can be accepted almost as is.

[Figure]

Thank you for the positive evaluation.

- What can be improved is a better link between the various genetic mechanisms discussed and a preferred solution. The validity of some of these mechanisms is not really fully clear in the manuscript. For instance, I would see lithospheric folding as fairly suitable mechanism providing an advanced explanation. However, the authors discard this mechanism because "a region that was subsiding until the onset of inversion will not become uplifted but exhibit accelerated subsidence under tangential compression", which is an unclear argument. This is either not well explained or incorrect: sure that subsidence may be enhanced by lithospheric folding in basins, we see such effects in many worldwide places. In a similar way, other potential mechanisms are not fully clear in the manuscript, at least to me. Therefore, to increase the impact of the paper, I suggest to revise, explain better and be more quantitative to all mechanisms explained in Section 7. Otherwise, as said above, this is a very nice contribution that fits perfectly the scope of the journal.

Yes, we agree that a well-elaborated solution that fully explains our findings would be desirable. Given the length and scope of the paper as presented now, we decided to discuss first-order estimates of some (more or less) possible mechanisms as endmember scenarios. This helps in roughly evaluating if they may account for the observed size, magnitudes and rates of uplift. Even such simple approach proves some mechanisms possible or partly possible, others impossible. A more detailed evaluation of the specific mechanisms and combinations thereof is left for follow-up studies. Regarding lithospheric folding, we have strengthened our reasoning in the revised text that this mechanism cannot be considered a main cause of regional doming. The large-scale structure indicates that the area that underwent regional doming coincides with a wide syncline today (see Fig. 11). Before the uplift event this syncline must have been deeper. This decrease in fold amplitude accompanying uplift cannot be the result of maintained or increased horizontal stress. It could be due to a decrease in stress if we assume that the syncline was formed or tightened by lithospheric folding (cf. Nielsen

et al., 2005). However, since stress relaxation cannot exhume the syncline more than it was originally deepened by horizontal stress, this assumption restricts the time available for deposition of the missing overburden to the short interval of the inversion phase (approx. 90 to 75 Ma). This is considered a highly unlikely scenario.
* * *

---

## Editor Decision (ED1)

**Late Cretaceous to Paleogene exhumation in Central Europe – localized inversion vs. large-scale domal uplift**

Hilmar von Eynatten[1], Jonas Kley[2], István Dunkl[1], Veit-Enno Hoffmann[1], Annemarie Simon[1]

[1]University of Göttingen, Geoscience Center, Department of Sedimentology and Environmental Geology,
Goldschmidtstrasse 3, 37077 Göttingen, Germany
[2]University of Göttingen, Geoscience Center, Department of Structural Geology and Geodynamics,
Goldschmidtstrasse 3, 37077 Göttingen, Germany

*Correspondence to*: Hilmar von Eynatten (heynatt@gwdg.de)

**Abstract.** Large parts of Central Europe have experienced exhumation in Late Cretaceous to Paleogene time. Previous studies mainly focused on thrusted basement uplifts to unravel magnitude, processes and timing of exhumation. This study provides, for the first time, a comprehensive thermochronological dataset from mostly Permo-Triassic strata exposed adjacent to and between the basement uplifts in central Germany, comprising an area of at least some 250-300 km across. Results of apatite fission track and (U-Th)/He analyses on >100 new samples reveal that (i) km-scale exhumation affected the entire region, (ii) thrusting of basement blocks like the Harz Mountains and the Thuringian Forest focused in the Late Cretaceous (about 90-70 Ma) while superimposed domal uplift of central Germany is slightly younger (about 75-55 Ma), and (iii) large parts of the domal uplift experienced removal of 3 to 4 km of Mesozoic strata. Using spatial extent, magnitude and timing as constraints suggests that thrusting and crustal thickening alone can account for no more than half of the domal uplift. Most likely, dynamic topography caused by upwelling asthenosphere has contributed significantly to the observed pattern of exhumation in central Germany.

**1 Introduction**

Widespread intraplate compressional stresses affected Central Europe in Cretaceous to Paleogene time and generated numerous basement uplifts and inverted sedimentary basins (e.g. Ziegler et al., 1995; Kley and Voigt, 2008). The basement uplifts cover a large area of at least 1300 km west to east and 600 km north to south extension. It stretches from the Ardennes in Belgium (western Rhenish Massif) to south-eastern Poland (Holy Cross Mountains) and includes prominent fault-bounded blocks composed of crystalline basement rocks and pre-Permian metasedimentary rocks such as the Bohemian Massif, the Vosges and Black Forest, and the Harz Mountains (Fig. 1A). The major phase of exhumation and uplift is mostly assigned to the Late Cretaceous (Kley and Voigt, 2008). However, earlier onset of exhumation and uplift and/or its continuation into the Paleogene are proposed for certain areas and structures (e.g. Barbarand et al., 2018; Sobczyk et al., 2020).

[Figure]

**A**

54°N
Hamburg
FH
NEGB
Berlin
Warsaw
LSB
52°N
MB
H
Bruxelles
TB
K   S
TF
Prague
50°N
AR
Bohemian Massif
Carpathians
OW
N
URG
km
48°N
6°E   Basel   10°E   14°E Alps   18°E

| | |
|---|---|
| Upper Cretaceous | Variscan basement & Lower Permian |
| Lower Cretaceous | |
| Jurassic | major reverse fault |
| Upper Permian & Triassic | |

**B**

NNE
Flechtingen High (FH)
Harz Mts. (HM)
Thuringian Forest (TF)
SSW

ca. 100 km

**C**

54°N

78-77(a)
75-72(b)
52°N
83-73(c)
108-74(d)   102-50(i)
202-88(k)
280-136(e)
81-69(g)   210-45(h)   121-39(j)
105-70(q)
50°N
324-161(n)
290-130(f)
110-54(p)
152-44(l)
103-15(r)
148-83(o)
48°N
233-92(m)
6°E   10°E   14°E   18°E

[revised manuscript text omitted]

---

## Author Response (AR2)

Response to comments by the executive editor (**Publish subject to technical corrections**)

*We are pleased to let you know that your manuscript is accepted for publication in the special issue on Inversion tectonics in Solid Earth, pending technical corrections. Please find the required corrections attached as edits and comments on the manuscript file. Thank you for submitting your work with us.*

*With best wishes,*
*Susanne Buiter*

Dear colleague,

we have corrected the text accordingly (mostly upper/lower case issues related to the use of Late/Early and Lower/Upper). We have added a small inset to Figure 1 and mentioned this in the respective caption. We would like to keep figures 4, 11, and 12 in landscape orientation because we think otherwise too much detail would be lost.

Best Wishes,

Hilmar v. Eynatten